# TCR meta-clonotypes for biomarker discovery with *tcrdist3* enabled identification of public, HLA-restricted clusters of SARS-CoV-2 TCRs

Koshlan Mayer-Blackwell[1], Stefan Schattgen[2], Liel Cohen-Lavi[3], Jeremy C Crawford[4], Aisha Souquette[5], Jessica A Gaevert[4], Tomer Hertz[6], Paul G Thomas[5], Philip Bradley[7], Andrew Fiore-Gartland[1]*

[1]Vaccine and Infectious Disease Division, Fred Hutchinson Cancer Research Center, Seattle, United States; [2]Department of Immunology, St Jude Children's Research Hospital, Memphis, United States; [3]Department of Industrial Engineering and Management, Ben-Gurion University of the Negev, Be'er Sheva, Israel; [4]Department of Immunology, St Jude Children's Research Hospital, Memphis, United States; [5]St Jude Children's Research Hospital, Memphis, United States; [6]Shraga Segal Department of Microbiology and Immunology, Ben-Gurion University of the Negev, Be'er Sheva, United States; [7]Fred Hutchinson Cancer Research Center, Seattle, United States

*For correspondence:
agartlan@fredhutch.org

**Abstract** T-cell receptors (TCRs) encode clinically valuable information that reflects prior antigen exposure and potential future response. However, despite advances in deep repertoire sequencing, enormous TCR diversity complicates the use of TCR clonotypes as clinical biomarkers. We propose a new framework that leverages experimentally inferred antigen-associated TCRs to form meta-clonotypes – groups of biochemically similar TCRs – that can be used to robustly quantify functionally similar TCRs in bulk repertoires across individuals. We apply the framework to TCR data from COVID-19 patients, generating 1831 public TCR meta-clonotypes from the SARS-CoV-2 antigen-associated TCRs that have strong evidence of restriction to patients with a specific human leukocyte antigen (HLA) genotype. Applied to independent cohorts, meta-clonotypes targeting these specific epitopes were more frequently detected in bulk repertoires compared to exact amino acid matches, and 59.7% (1093/1831) were more abundant among COVID-19 patients that expressed the putative restricting HLA allele (false discovery rate [FDR]<0.01), demonstrating the potential utility of meta-clonotypes as antigen-specific features for biomarker development. To enable further applications, we developed an open-source software package, *tcrdist3*, that implements this framework and facilitates flexible workflows for distance-based TCR repertoire analysis.

## Editor's evaluation

This paper introduces and validates a novel concept which will be of great interest to all those interested in T cell immunity and especially the T cell receptor repertoire. The concept builds on the idea that TCRs to the same antigen often share sequence similarities, which they quantify using a bespoke tool tcrdist3. Using this tool they develop the idea of a meta-clone, a set of TCRs sharing biochemical similarities and potentially recognising the same antigen. In this paper they further show that such clonotypes may show increased sharing between HLA-related individuals, and explore

the use of such clonotypes in characterising antigen-specific immune response across cohorts of individuals.

## Introduction

An individual's unique repertoire of T-cell receptors (TCRs) is shaped by antigen exposure and is a critical component of immunological memory (*Emerson et al., 2017*; *Welsh and Selin, 2002*). With the advancement of immune repertoire profiling, TCR repertoires are a largely untapped source of biomarkers that could potentially be used to predict immune responses to a wide range of exposures including viral infections (*Wolf et al., 2018*), tumor neoantigens (*Ahmadzadeh et al., 2019*; *Chiou et al., 2021*; *Kato et al., 2018*), or environmental allergens (*Cao et al., 2020*). However, the extreme diversity characterizing TCR repertoires, both within and between individuals, presents major hurdles to biomarker development. Using peptide—major histocompatibility complex (pMHC) tetramer sorting to focus on TCRs recognizing individual epitopes, which depends on knowing the peptide antigen and its MHC restriction, typically reveals that many distinct TCRs are able to recognize even a single pMHC (*Coles et al., 2020*; *Meysman et al., 2019*). This complicates detection of population-wide signatures of antigen exposure. Modeling (*Elhanati et al., 2018*) and empirical evidence (*Soto et al., 2019*) suggest that only 10–15% of single-chain TCRs are public or shared by multiple individuals. Furthermore, only a fraction of the repertoire can be sampled, making it difficult to reproducibly detect relevant TCR clonotypes from an individual, let alone reliably detect public clonotypes in a population; in practice, the problem can be exacerbated by heterogeneous repertoire sequencing depth, which affects the precision with which the frequency of rare TCRs can be estimated. Thus, individual T-cell clonotypes are currently suboptimal and underpowered for population-level investigations of TCR specificity, which limits their application in the development of TCR-based clinical biomarkers.

In this study, we describe a framework for engineering 'meta-clonotypes': groupings of TCRs sharing biochemically similar complementarity determining regions (CDRs), which enable population-level biomarker development (*Figure 1*). Previously, we introduced TCRdist, a biochemically informed distance metric that enabled grouping of paired αβ TCRs by antigen specificity based on their sequence similarity (*Dash et al., 2017*). TCRdist is correlated with edit distance, but it can vary considerably among TCRs with identical edit distances (*Figure 2*). While other tools exist to identify statistically anomalous groups of TCRs within a single sample that may be indicative of a polyclonal response to antigenic selection (*Glanville et al., 2017*; *Huang et al., 2020*; *Pogorelyy et al., 2019*; *Pogorelyy and Shugay, 2019*; *Ritvo et al., 2018*; *Shugay et al., 2015*), the meta-clonotype framework has been developed for a different task: leveraging receptor–antigen associations determined from in vitro experiments to create public, antigen-associated meta-clonotypes from otherwise private TCRs. This application is made possible by a new open-source Python3 software package *tcrdist3* that brings flexibility to distance-based repertoire analysis, allowing customization of the distance metric, and at-scale computation with sparse data representations and parallelized, byte-compiled code.

The framework is based on TCR sequences that have been experimentally enriched for antigen recognition, most commonly by sorting T cells labeled by peptide–MHC multimers or by activation-induced markers upon stimulation (we refer to these as 'antigen-associated' TCRs). Each meta-clonotype is defined by an antigen-associated centroid TCR and a TCRdist radius chosen so that the expected frequency of antigen-naive receptors within the radius is low. A CDR3 'motif' is constructed from the subset of antigen-associated TCRs within the radius to further refine the meta-clonotype's specificity. Together the centroid receptor, radius, and the CDR3 motif can be used to search for conformant TCRs in large bulk-sequenced repertoires and quantify their frequency (*Figure 1*). As intended, we find that TCR centroids, which are often private, gain publicity as meta-clonotypes. The expanded publicity of meta-clonotypes provides an opportunity to develop population-level biomarkers that may depend on antigen-specific features of the TCR repertoire. Shifting the focus of repertoire analysis from clonotypes to meta-clonotypes increases statistical power; grouping similar clonotypes reduces the sparsity of finite repertoire samples and increases the precision with which antigen-specific cell abundance can be estimated.

To demonstrate one potential application of meta-clonotypes and to characterize their ability to estimate the frequency of similar antigen-specific T cells in bulk-sequenced TCR repertoires, we apply

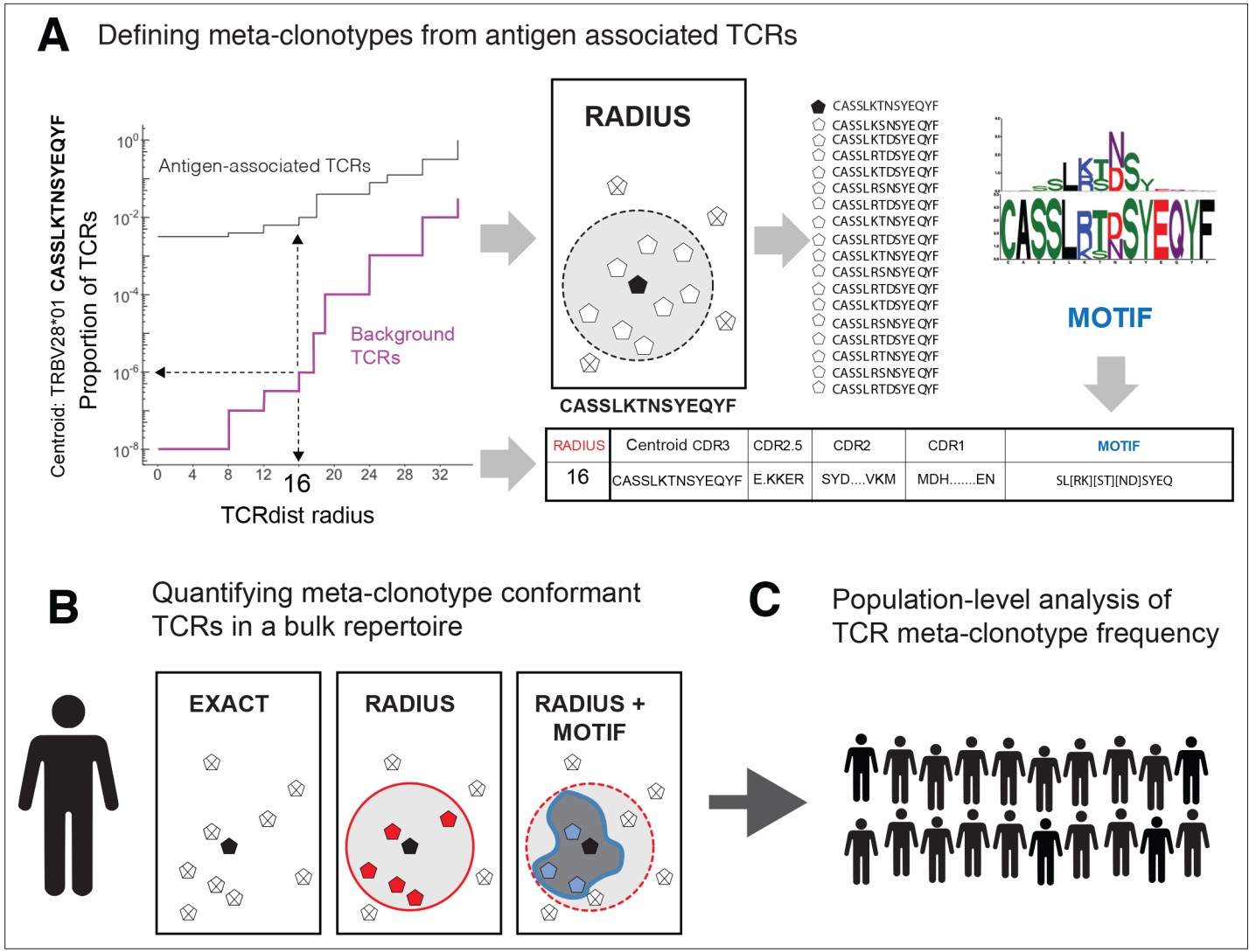

**Figure 1.** T-cell receptor (TCR) meta-clonotypes. (**A**) *Defining meta-clonotypes from antigen-associated TCRs*. Sets of antigen-associated TCRs were used together with synthetic background repertoires to engineer TCR meta-clonotypes that define biochemically similar TCRs based on a centroid TCR and a TCRdist radius. For each antigen-specific clonotype, we used *tcrdist3* to evaluate the proportion of TCRs spanned at different TCRdist radii within (i) its antigen-associated TCR set (black) and (ii) a synthetic control V- and J-gene-matched background set (purple). A synthetic background was generated using 100,000 Optimized Likelihood estimate of Immunoglobulin Amino acid sequences (OLGA)-generated TCRs and 100,000 TCRs subsampled from umbilical cord blood; OLGA-generated TCRs were sampled to match the V–J gene frequency in each MIRA receptor set, with weighting to account for the sampling bias (see Methods for details). The objective was to select the largest radius that includes no more than an estimated proportion of $1E^{-6}$ TCRs in the background. The subset of antigen-associated TCRs spanned by the selected radius were then used to develop an additional meta-clonotype motif constraint based on conserved residues in the complementarity determining region (CDR)3 (see Methods for details). An example logo plot shows the CDR3 β-chain motif formed from TCRs – activated by a SARS-CoV-2 peptide (MIRA55 ORF1ab amino acids 1316:1330, ALRKVPTDNYITTY) – within a TCRdist radius 16 of this meta-clonotype's centroid TCR. (**B**) *Quantifying meta-clonotype conformant TCRs in bulk repertoires*. The definition of each TCR meta-clonotype can be used to quantify the frequency of similar TCRs in bulk repertoires. EXACT sequences match the meta-clonotype centroid at the amino acid level, RADIUS-conformant sequences diverge from the centroid by no more than the radius distance, and RADIUS + MOTIF conformant sequences is the subset of radius-conformant TCRs with a CDR3 sequences matching the meta-clonotype's CDR3 motif. (**C**) *Population-level analysis of TCR meta-clonotype frequency*. The frequency of meta-clonotype conformant sequences in multiple bulk repertoires allows comparison across a population. In this study, to test whether meta-clonotypes carry important antigen-specific signals above and beyond individual clonotypes, we searched for meta-clonotype conformant TCRs in COVID-19 patients with repertoires collected 0–30 days after diagnosis. We found stronger associations with predicted HLA restrictions based on counts of meta-clonotype conforming TCRs compared to associations using counts of exact clonotypes.

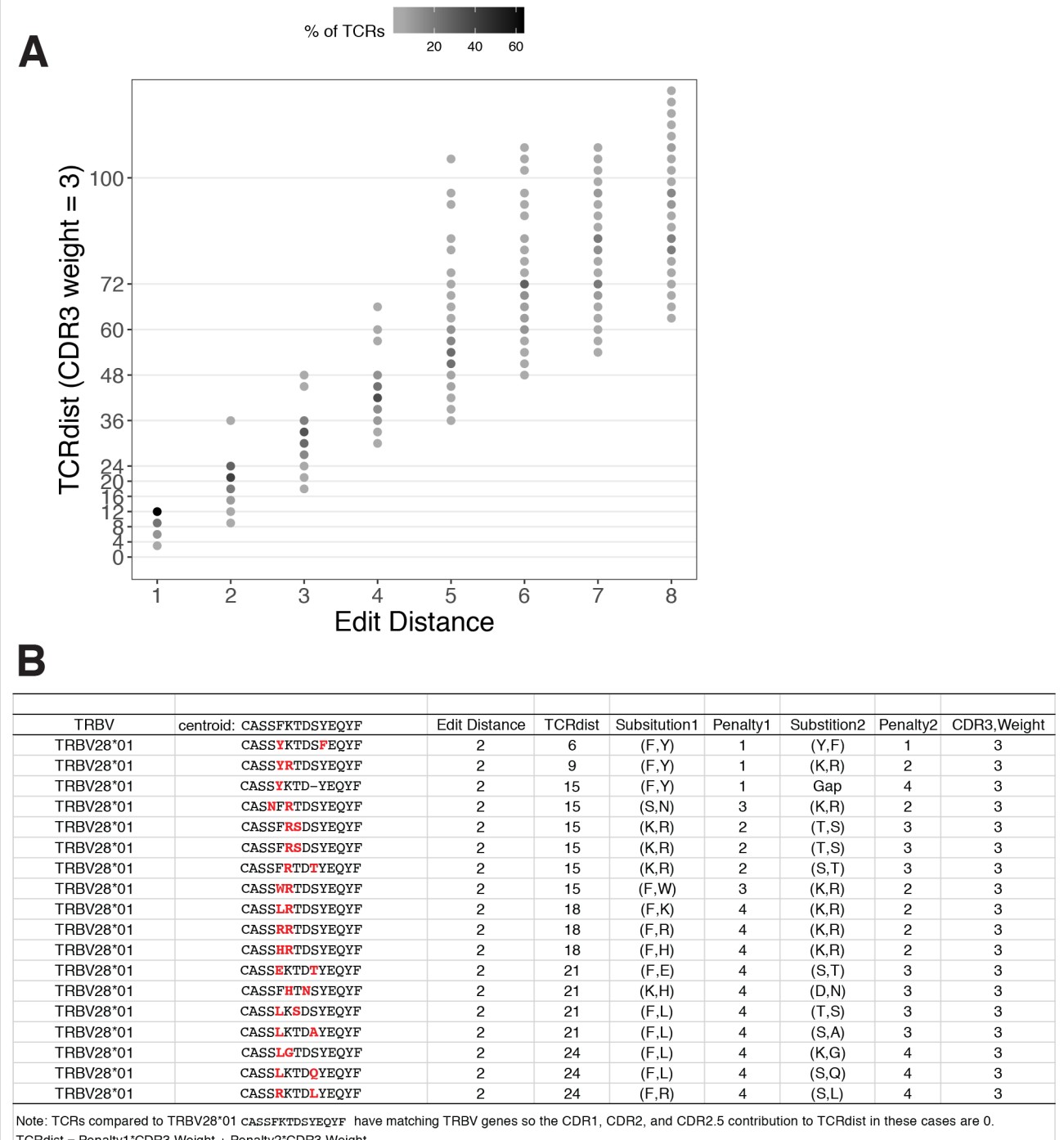

**Figure 2.** TCRdist compared to edit distance. (**A**) Correspondence between edit distance (*x*-axis) and TCRdist (*y*-axis) for MIRA55 T-cell receptors (TCRs) with matching TRBV genes. The grayscale colormap shows the percentage of TCRs with a given TCRdist score within each edit distance category. (**B**) Examples of complementarity determining region (CDR)3s with TCRdist varying between 6 and 24 units among sequences with edit distance 2 (2 substitutions) from a centroid with matching TRBV genes. TCR distances range based on differential penalties assigned to specific residue substitutions.

the meta-clonotype framework to a large publicly available dataset of SARS-CoV-2 antigen-associated TCRs. The dataset comes from a recent study that sought to elucidate the role of cellular immune responses in acute SARS-CoV-2 infection and examined the TCR repertoires of patients diagnosed with COVID-19 disease. Researchers used an assay based on antigen stimulation and flow cytometric sorting of activated CD8+ T cells to identify SARS-CoV-2 peptide-associated TCR β-chains; the assay

is called 'multiplex identification of TCR antigen specificity' or MIRA (*Klinger et al., 2015*) and the output is a set of predicted antigen-associated TCR sequences. Data from these experiments were released publicly in July 2020 by Adaptive Biotechnologies and Microsoft as part of 'immuneRACE' and their efforts to stimulate science on COVID-19 (*Nolan et al., 2020*; *Snyder et al., 2020*). The MIRA antigen stimulation assays identified 253 sets of 6 or more TCR β-chains associated with CD8+ T cells activated by exposure to SARS-CoV-2 peptides, with TCR sets analyzed ranging in size from 6 to 16,607 TCRs (*Supplementary file 1b*); we refer to these sets as MIRA0 through MIRA252 in rank order by their size. The deposited immuneRACE datasets also included bulk TCR β-chain repertoires from 694 patients within 0–30 days of COVID-19 diagnosis. Our analysis of these data demonstrates how it is possible to define public meta-clonotypes from sets of private antigen-associated TCRs and directly evaluates their ability to carry population-level antigen-specific signals in comparison with individual clonotypes.

## Results

### Experimental enrichment of antigen-specific T cells allows discovery of TCRs with biochemically similar neighbors

Searching for identical TCRs within a repertoire – arising either from clonal expansion or convergent nucleotide encoding of amino acids in the CDR3 – is a common strategy for identifying functionally important receptors. However, in the absence of experimental enrichment procedures, observing T cells with the same amino acid TCR sequence in a bulk sample is rare. For example, in 10,000 β-chain TCRs from an umbilical cord blood sample, less than 1 % of TCR amino acid sequences were observed more than once, inclusive of possible clonal expansions (*Figure 3A*). By contrast, a valuable feature of antigen-associated TCRs is the presence of multiple receptors with identical or highly similar amino acid sequences (*Figure 3A*). For instance, 45% of amino acid TCR sequences were observed more than once (excluding clonal expansions) in a set of influenza M1(GILGFVFTL)-A*02:01 peptide–MHC tetramer-sorted subrepertoires from 15 subjects (*Dash et al., 2017*). Enrichment was evident compared to cord blood for additional peptide–MHC tetramer-sorted subrepertoires obtained from VDJdb (*Shugay et al., 2018*), though the proportion of TCRs with an identical or similar TCR in each set was heterogeneous.

We investigated the degree to which the MIRA assay employed by *Nolan et al., 2020* identified TCRs with identical or similar amino acid sequences. In general, across sets of MIRA-identified β-chain TCRs, each associated with a different antigen, the proportion of amino acid sequences observed more than once was generally lower than in the tetramer-enriched repertoires and varied considerably across the sets; some MIRA sets resembled tetramer-sorted subrepertoires (*Figure 3B*; see MIRA133), while others were more similar to unenriched repertoires (*Figure 3B*; see MIRA90). The increased diversity in MIRA-enriched TCR sets versus tetramer-enriched TCR sets may, in part, be explained by: (1) peptides being presented by the full complement of the native host's MHC molecules compared to a single defined peptide–MHC complex, (2) recruitment of lower affinity receptors, (3) antigen specificity conferred primarily by the alpha rather than the sequenced beta chain, or (4) nonspecific 'bystander' activation in the MIRA stimulation assay. From an experimental standpoint, MIRA offers the benefit of being able to identify TCRs associated with an antigen before a specific pMHC has been identified; however, the resultant diversity in antigen-associated TCRs recovered by MIRA poses a challenge for identifying relevant TCR motifs associated with multiple possible TCR:pMHC interactions.

### TCR biochemical neighborhood density is heterogeneous among set of antigen-associated TCRs

We next investigated the proportion of unique TCRs with at least one biochemically similar neighbor among TCRs with the same putative antigen specificity. We and others have shown that a single peptide–MHC epitope is often recognized by many distinct TCRs with closely related amino acid sequences *Dash et al., 2017*; in fact, the detection of such clusters in bulk-sequenced repertoires is the basis of several existing tools: GLIPH (*Glanville et al., 2017*; *Huang et al., 2020*), ALICE (*Pogorelyy et al., 2019*), TCRNET (*Ritvo et al., 2018*), and RepAn (*Yohannes et al., 2021*). Therefore, to better understand sets of antigen-associated TCRs, like the SARS-CoV-2 MIRA data, we evaluated the

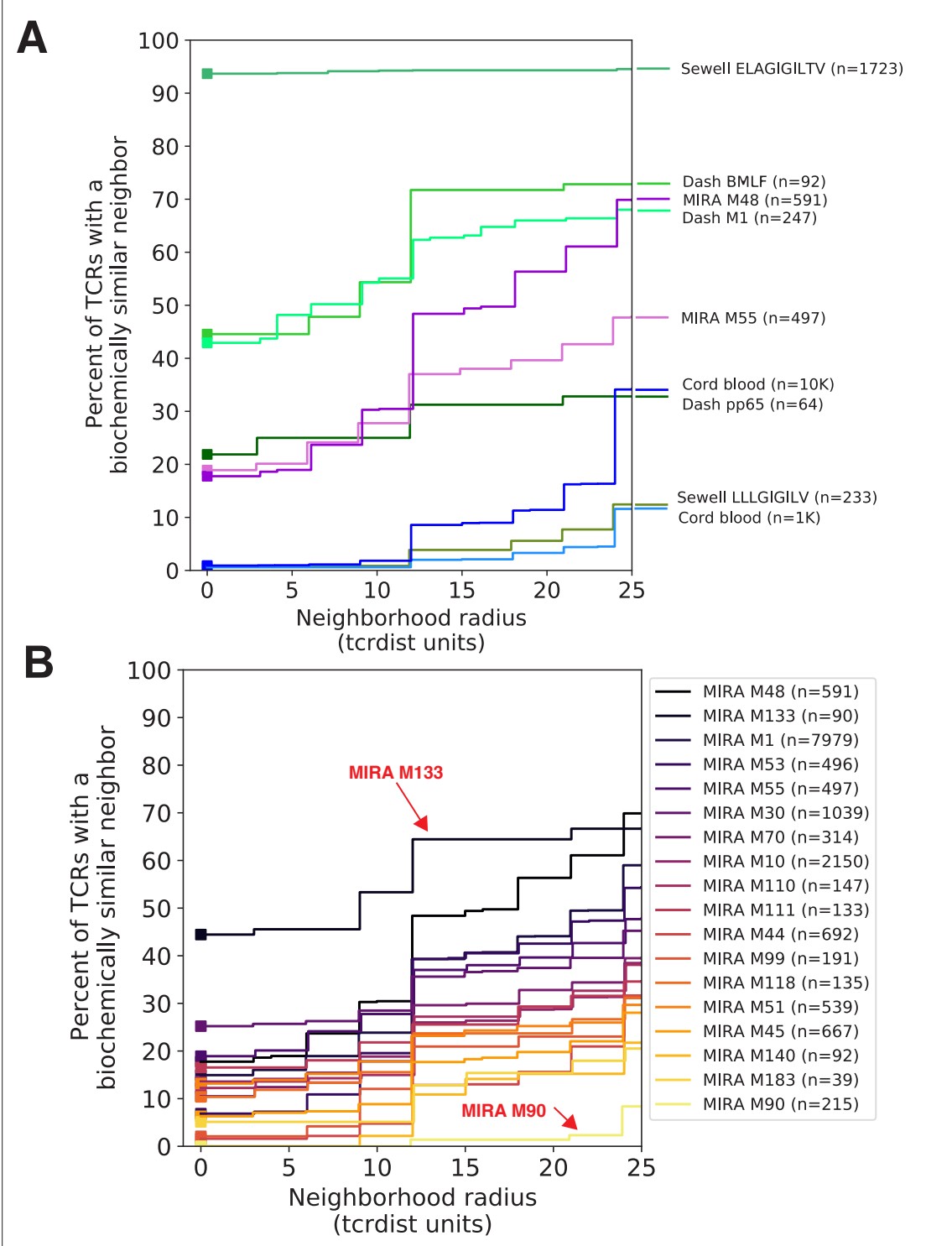

**Figure 3.** Experimental enrichment of antigen-associated T-cell receptors (TCRs) increases neighbor density. (**A**) TCR repertoire subsets obtained by single-cell sorting with peptide–major histocompatibility complex (MHC) tetramers (green), MIRA peptide stimulation enrichment (MIRA55, MIRA48; purple), or random subsampling of umbilical cord blood (1000 or 10,000 TCRs; blue). Biochemical distances were computed among all pairs of TCRs in each subset using the TCRdist metric. Neighborhoods were formed around each TCR using a variable radius (x-axis) and the percent of TCRs in the set with at least one other TCR within its neighborhood was computed; notably the line represents a summary of TCRs in each set and is therefore more precise for larger TCR sets. A radius of zero indicates the proportion of TCRs that have at least one TCR with an identical amino acid sequence (solid square). Dash BMLF (Epstein–Barr Virus), M1 (Influenza), and pp65 (Cytomegalovirus) refer to epitopes from ***Dash et al., 2017***. ELAGIGILTV (Human Mart-1 antigen) and LLLGIFILV (HM1.24 antigen in multiple myeloma) downloaded from VDJdb (***Shugay et al., 2018***), which were submitted by Andrew

*Figure 3 continued on next page*

Figure 3 continued

Sewell et al. (**B**) Analysis of MIRA sets for which the participants contributing the TCRs were significantly enriched with a specific class I HLA allele **Supplementary file 1c**. Colors are assigned based on the vertical ranking of the lines along the right *y*-axis and match the order in the color legend.

neighborhood surrounding each TCR, defined as the set of similar TCRs whose sequence divergence is within a specified radius. The radius was measured using TCRdist, a position weighted, multi-CDR distance metric. Briefly, differences in the amino acid sequences of the CDRs are totaled based on the number of gaps (−4) and their BLOSUM62 substitution penalties (ranging from 0 to −4) with a default threefold weighting on CDR3 substitutions (see Methods for details of *tcrdist3* reimplementation of TCRdist); a one amino acid mismatch in the CDR3 results in a maximal distance of 12 TCRdist units (tdus). As the radius about a TCR centroid expands, the number of TCRs it encompasses naturally increases. The increase was greater among the sets of antigen-associated TCRs compared to the 'background' repertoires that were not experimentally enriched for antigen-specific T cells (*Figure 3*).

To better understand the relationship between the TCR distance radius and the density of proximal TCRs, we constructed empirical cumulative distribution functions (ECDFs) for each unique TCR (*Figure 4*). The ECDF for each unique TCR (each represented by one line in *Figure 4*) shows the proportion of all TCRs within the indicated radius; those with sparse neighborhoods appear as lines that remain low and do not increase along the *y*-axis even as the search radius expands (lines are hidden by the *x*-axis). The proportion of these TCRs with sparse or empty neighborhoods (ECDF proportion <0.001) is indicated by the height of the gray area plotted below the ECDF (*Figure 4*). We observed the highest density neighborhoods within repertoires that were sorted based on binding to a single peptide–MHC tetramer. For instance, with the influenza M1(GILGFVFTL)-A*02:01 tetramer-enriched repertoire from 15 subjects, we observed that many TCRs were concentrated in dense neighborhoods, which included as much as 30 % of the other influenza M1-recognizing TCRs within a radius of 12 tdus (*Figure 4A*). Notably there were also many TCRs with empty or sparse neighborhoods using a radius of 12 tdus (111/247, 44%) or 24 tdus (83/247, 34%). Based on previous work (*Dash et al., 2017*), we assume that the majority of these tetramer-sorted CD8+ T cells with few proximal neighbors do indeed bind the influenza M1:A*02:01 tetramer. This suggests that TCRs within sparse neighborhoods represent uncommon modes of antigen recognition and highlights the broad heterogeneity of neighborhood densities even among TCRs recognizing a single peptide–MHC.

Neighbor densities for individual TCRs within the MIRA sets were highly heterogeneous. Densities for an illustrative MIRA set are shown in *Figure 5* (MIRA55:ORF1ab; 1316:1330 [amino acid]; peptide ALRKVPTDNYITTY). Within this antigen-associated repertoire, at 24 tdus 8.9 % (44/497) of TCR neighborhoods included >10% of the other antigen-activated CD8+ TCRs (*Figure 5A*). As expected, TCR neighborhoods in the umbilical cord blood repertoire were sparser (*Figure 5B*); the densest neighborhood included only 0.13 % of the repertoire at 24 tdus. We also noted that TCRs with sparse or empty neighborhoods tended to have longer CDR3 loops (*Figure 4C*) and lower generation of probability ($p_{gen}$; *Figure 5B*). This is consistent with mathematical modeling that shows that TCRs with longer CDR3 loops have a lower $p_{gen}$ during genomic recombination of the TCR locus (*Marcou et al., 2018*; *Murugan et al., 2012*; *Sethna et al., 2019*). Absent strong selection for antigen recognition, longer TCRs with lower generation probabilities are thus likely to have less dense biochemical neighborhoods. Together, these observations suggest that biochemical neighborhood density is highly heterogeneous among TCRs and that it may depend on mechanisms of antigen recognition as well as receptor V(D)J recombination biases (*Thomas and Crawford, 2019*).

## Meta-clonotype radius can be tuned to balance sensitivity and specificity

The utility of a TCR-based biomarker depends on the antigen specificity of the TCRs. Therefore, a limitation of distance-based clustering is the presence of similar TCR sequences that lack the ability to recognize the target antigen. To be useful, a meta-clonotype definition should be broad enough to capture multiple biochemically similar TCRs with shared antigen recognition, but not excessively broad as to include a high proportion of nonspecific TCRs. Statistically, we think of a meta-clonotype definition as a way to balance sensitivity and specificity, respectively, the ability to include antigen-recognizing TCRs and exclude nonspecific TCRs. Because the density of neighborhoods around TCRs are heterogeneous, we hypothesized that the optimal radius defining a meta-clonotype may differ for

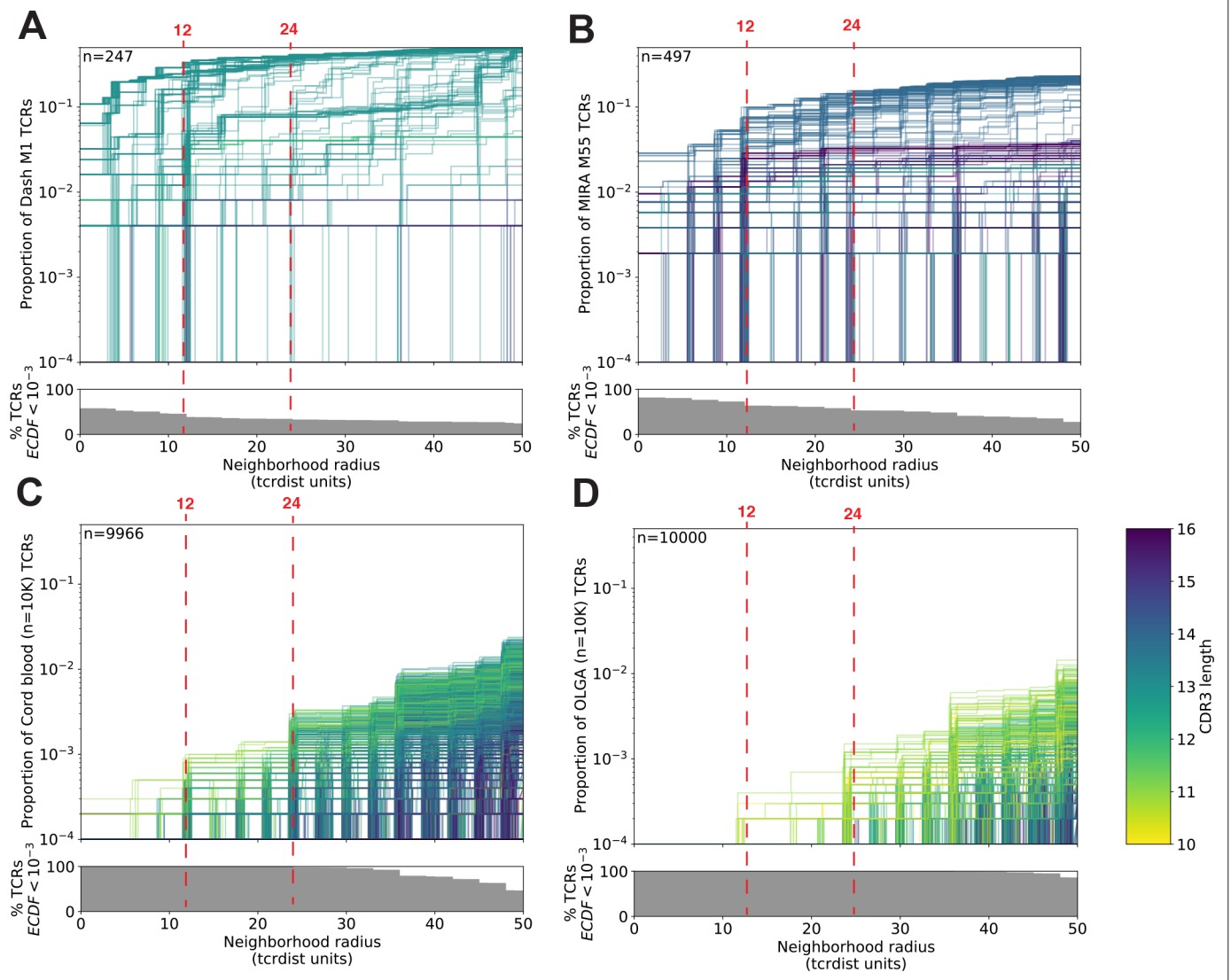

**Figure 4.** T-cell receptor (TCR) neighborhoods have higher density among TCRs that have been experimentally enriched for antigen-specific T cells compare to unenriched repertoires. TCR β-chains from (**A**) a peptide–major histocompatibility complex (MHC) tetramer-enriched subrepertoire (n = 247), (**B**) a MIRA peptide stimulation-enriched subrepertoire (n = 497), or (**C**) an umbilical cord blood unenriched repertoire (n = 9966), and (**D**) synthetically generated sequences using Optimized Likelihood estimate of Immunoglobulin Amino acid sequences (OLGA; n = 10,000; *Sethna et al., 2019*). Within each subrepertoire, an empirical cumulative distribution function (ECDF) was estimated for each TCR (one line) acting as the centroid of a neighborhood over a range of distance radii (x-axis). Each ECDF shows the proportion of TCRs within the set with a distance to the centroid less than the indicated radius. ECDF color corresponds to the length of the complementarity determining region (CDR)3-β loop. ECDF curves were randomly shifted by <1 unit along the x-axis to reduce overplotting. Vertical ECDF lines starting at $10^{-4}$ indicate no similar TCRs at or below that radius. Percentage of TCRs with an ECDF proportion <$10^{-3}$ (bottom panels), indicates the percentage of TCRs without, or with very few biochemically similar neighbors at the given radius.

each TCR. To find the ideal radius we proposed comparing the relative density of a radius-defined neighborhood within a set of antigen-associated TCRs (*Figure 5A*) to the density of the radius-defined neighborhood within a background TCR repertoire (*Figure 5B,C*); here, the background repertoire can be any set of TCRs from antigen-naive repertoires. Though ideally this background comparator would explicitly exclude antigen-specific TCRs, we can use a nonselected repertoire as a background because the frequency of antigen-specific T cells in a large background drawn from antigen-naive donors is assumed to be low. Also, a nonselected background is a relevant comparator because it provides an estimate of the number of false detections we expect when each meta-clonotype is ultimately used to search for and quantify putatively antigen-specific sequences in bulk repertoires.

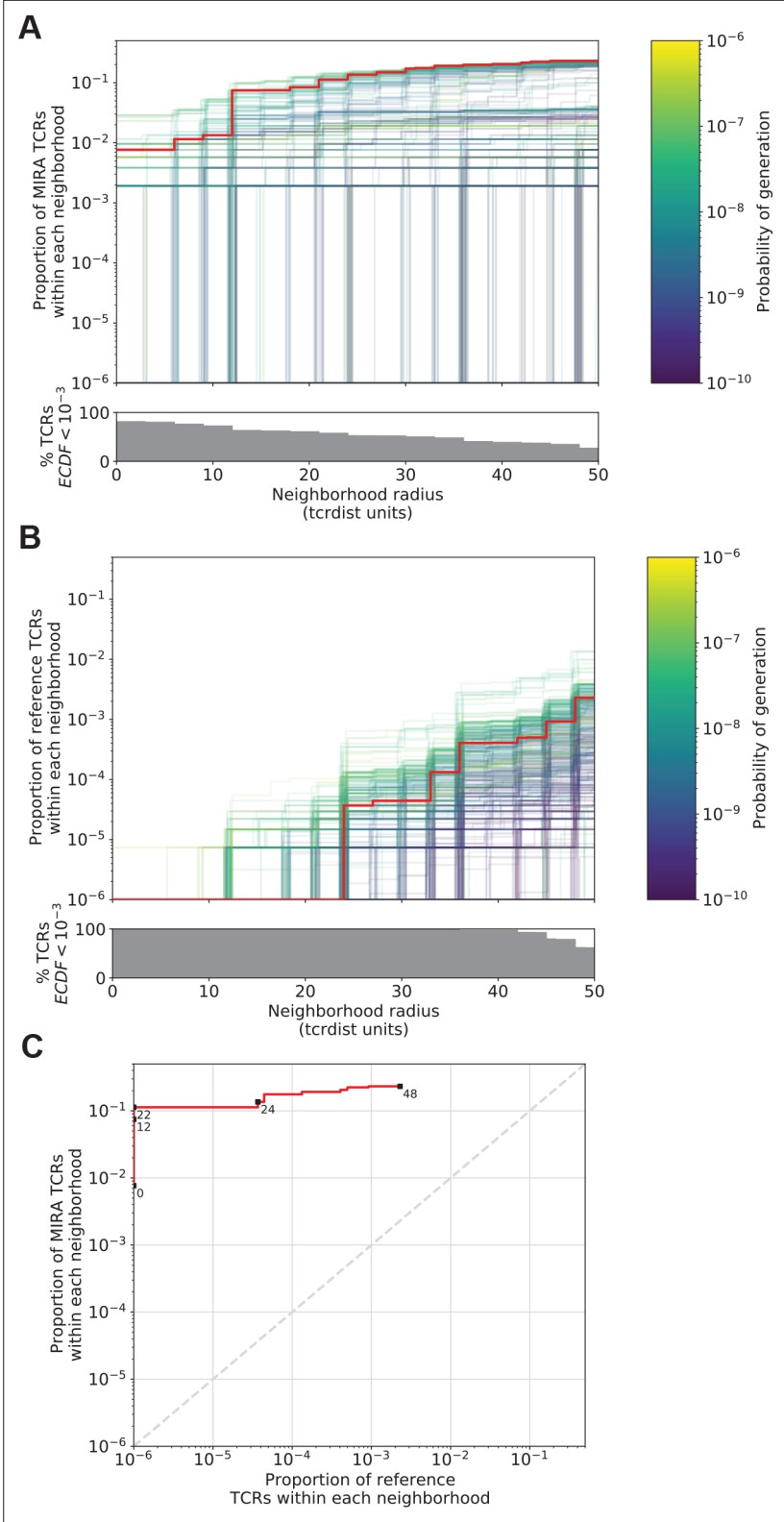

**Figure 5.** Radius-defined neighborhood densities within an antigen-associated and a synthetic background repertoire. (**A**) Each T-cell receptor (TCR) (one line, *n* = 497) in the MIRA55 antigen-associated set acts as the centroid of a neighborhood and an empirical cumulative distribution function (ECDF) is estimated over a range of distance radii (*x*-axis). Each ECDF shows the proportion of TCRs within the MIRA set having a distance to the centroid less than the indicated radius. The ECDF line color corresponds to the TCR probability of generation (p_gen)

*Figure 5 continued on next page*

*Figure 5 continued*

estimated using Optimized Likelihood estimate of Immunoglobulin Amino acid sequences (OLGA; *Sethna et al.,* ***2019***). The ECDF curves are randomly shifted by <1 unit along the *x*-axis to reduce overplotting. The bottom panel shows the percentage of TCRs with an ECDF proportion $<10^{-3}$. (**B**) Estimated ECDF for each MIRA55 TCR based on the proportion of TCRs in a synthetic background repertoire that are within the indicated radius (*x*-axis). A synthetic background was generated using 100,000 OLGA-generated TCRs and 100,000 TCRs subsampled from umbilical cord blood; OLGA-generated TCRs were sampled to match the V–J gene frequency in the MIRA 55 receptor set, with weighting to account for the sampling bias (see Methods for details). (**C**) Antigen-associated ECDF (*y*-axis) of one example TCR's neighborhood (red line) plotted against ECDF within the synthetic background (*x*-axis). Example TCR neighborhood is the same indicated by the red line in (**A**) and (**B**). The dashed gray line indicates neighborhoods that are equally dense with TCRs from the antigen-associated and background subrepertoires. Annotations indicate the meta-clonotype radius for each data point in TCRdist units.

An ideal radius would define a meta-clonotype with a high density of conformant sequences within a set of antigen-associated TCRs and a low density among a set of background TCRs. We demonstrate an algorithm for selecting an optimal radius for each TCR in the MIRA55:ORF1ab dataset, which includes TCRs from 15 COVID-19 diagnosed subjects (see Methods for details about MIRA and the immuneRACE dataset). First, an ECDF is constructed for each centroid TCR showing the relationship between the meta-clonotype radius and its sensitivity: its inclusion of similar antigen-recognizing TCRs, approximated by the proportion of TCRs in the antigen-associated MIRA set that are within the centroid's radius (*Figure 5A*). Next, an ECDF is constructed for each TCR showing the relationship between the meta-clonotype radius and its specificity: its exclusion of TCRs with divergent antigen recognition, approximated by the proportion of TCRs in a background repertoire within the centroid's radius (*Figure 5B*). The objective is to select the largest radius that includes no more than one in one-million background TCRs; while this threshold is arbitrary, practically it means that in a deeply sampled repertoire we expect to observe only a few TCRs within the radius and that deviation from this may indicate antigenic selection. Typically, to accurately estimate the frequency of a rare event, one would prefer to observe many such events and use the average; here, this would require having a background set of many millions of TCRs that would be used to evaluate every potential radius for each TCR centroid, presenting a substantial computational hurdle.

We noted that, based on germline encoded CDR residues alone, much of the TCR background is too distant from a single TCR to be relevant and therefore realized that efficiency could be gained by focusing on TCRs that share the same TRBV and TRBJ genes. Therefore, for each set of antigen-associated TCRs identified using MIRA, we created a two part background. One part consisted of 100,000 synthetic TCRs whose TRBV- and TRBJ-gene frequencies matched those of the antigen-associated TCRs; TCRs were generated using the software OLGA with slight modification to allow V–J gene directed sequence generation (*Marcou et al., 2018*; *Sethna et al., 2019*). The other part consisted of 100,000 umbilical cord blood TCRs sampled from 8 subjects (*Britanova et al., 2016*). This composition balanced denser sampling of sequences near the candidate meta-clonotype centroids with broad sampling of TCRs from an antigen-naive repertoire. The dense sampling of TCRs with similar V–J combinations to the antigen-associated TCRs allowed for estimation of the overall frequency of meta-clonotype neighbors in the background well below 1 in 200,000. Conceptually, this is achievable because we oversampled the TCRs that were more likely to be within the meta-clonotype radius, therefore greatly increasing the statistical efficiency and precision with which we could estimate the overall frequency of meta-clonotype neighbors in the background. This idea is an adaptation of methods that are commonly used to adjust survey results when the sampling has known biases (*Gelman, 2007*). It is helpful to demonstrate the concept with an example: suppose we generate background TCRs (i.e., OLGA-generated) from one V–J gene combination, if we find that 5/50,000 (i.e., $10^{-4}$) TCRs are within a TCR's radius, but that they are sampled from a V–J gene combination with 1% (i.e., $10^{-2}$) prevalence, the estimated frequency in the full background would be 1 in 1 million ($10^{-4} \times 10^{-2} = 10^{-6}$). Across all V–J gene defined strata in the synthetic background, the adjusted frequency $P_{BG}$ can be estimated as a weighted average, with V–J gene strata frequencies from the full background as the weights ($w_i$):

$$P_{BG} = \sum_{i=1}^{N_{VJ}} \frac{Y_i}{n_i} w_i$$

where $Y_i$ is the number of TCRs within the radius of the centroid in the $i$th V–J gene defined strata and $n_i$ is the number of total sampled TCRs in the strata. Estimating neighbor frequency from a V–J gene-matched background alone, however, assumes there are zero neighbors in the unobserved strata with nonmatched V–J genes. Therefore, to avoid overlooking other regions of TCR space, we combined the synthetic background with a uniformly sampled background from antigen-naive cord blood repertoires. This is potentially important because with TCRdist – in contrast to metrics that require a matched V-gene – it is possible to find biochemically similar TCRs with different V-genes.

We found that for each TCR, its radius-defined meta-clonotype was more abundant within a repertoire and more prevalent in a human cohort than the exact clonotype; for example, TCR meta-clonotypes formed from the MIRA55:ORF1ab TCR set were detected in 3–12 (median 6) of 15 HLA-A*01 participants in the MIRA cohort, despite 34 of the 46 centroid clonotype TCRs being private (i.e., found in only 1 of 15 HLA-A*01 participants) (*Figure 6*). Generally, the neighborhoods around TCR centroids with higher probabilities of generation consistently spanned a larger proportion of background TCRs across a range of radii, suggesting that a smaller radius may be desirable for forming meta-clonotypes from high $p_{gen}$ TCRs. With a large radius, most TCR centroids had high sensitivity but low specificity, indicated by the meta-clonotypes including both a high proportion of TCRs from the antigen-associated and background repertoires. Some TCRs had low sensitivity even at a radius of 24 tdus, which is indicative of a low $p_{gen}$ or a 'snowflake' TCR: a seemingly unique TCR among the antigen-associated and background TCRs. However, radius-defined neighborhoods around many TCRs in the MIRA55:ORF1ab repertoire included 1–10% of the antigen-associated TCRs (5–50 clonotypes) with a radius that included fewer than 0.0001 % of background TCRs (equivalent to 1 out of $10^6$), demonstrating a level of sensitivity and specificity that would be favorable for the development of a TCR biomarker. In *Supplementary file 1*, additional information is provided about the other MIRA sets and the properties of meta-clonotypes that were generated.

## Sensitivity of optimized meta-clonotype radius to background size and specification

We conducted sensitivity analyses to evaluate how the optimal radius was dependent upon the size of the synthetic background. Using meta-clonotype centroids from the MIRA55:ORF1ab TCR set we recomputed the optimal radius for each meta-clonotype using 2,000,000 background sequences (1,000,000 synthetics V–J-matched OLGA and 1,000,000 cord blood TCRs), then drew repeated random samples from that background varying in size from 25,000 to 1,000,000 TCRs. We found that with a smaller background the optimal radius tended to be greater and was more variable over repeat samples; this is consistent with a lower and more variable chance of finding a sequence in the background that would help constrain the optimal radius. As the size of the background increased to 1,000,000 the estimates of the optimal radius generally approached the estimate using a 2,000,000 TCR background (*Figure 7*). For large sets of antigen-associated TCRs (i.e., greater than 10,000) or for users with modest computing resources, using 200,000 sequences with V–J-matched sampling strikes a balance between computational efficiency and bias in radius estimates, where the median radius over many iterations coincided with the median radius estimated from a background of 2,000,000 TCRs. Generally, the potential bias in estimation of an optimal radius is small, with the IQR ranging from 0 to 6 tdus, however, reducing bias by estimating optimal radii from a larger synthetic background (1–2 million TCRs) may still be prudent and computationally tractable. For instance, with *tcrdist3,* computing radii for 500 unique MIRA TCRs using a synthetic background of 1,000,000 TCRs (500,000 synthetic V–J matched and 500,000 sampled from cord blood) can be completed in 1 min using 12 CPUs and 48 GB of memory.

Next, for a fixed background size (200,000 TCRs), we evaluated the ability to estimate the optimal radius using three background sets: (1) 100,000 V–J gene-matched OLGA + 100,000 cord blood (primary background), (2) 200,000 V–J gene-matched OLGA, or (3) 200,000 OLGA without V–J gene matching (*Figure 7*). Each method was benchmarked against the optimal radii estimated from 1,000,000 cord blood TCRs from eight individuals. We found that the 200,000 TCRs generated from OLGA without V–J gene-matched sampling resulted in radii estimates that were substantially larger and more variable than those that were estimated using V–J gene-matched sampling and adjustment; this bias suggests that the resulting meta-clonotypes would have reduced specificity and therefore supports use of V–J gene focused sampling and adjustment to obtain less biased and more efficient

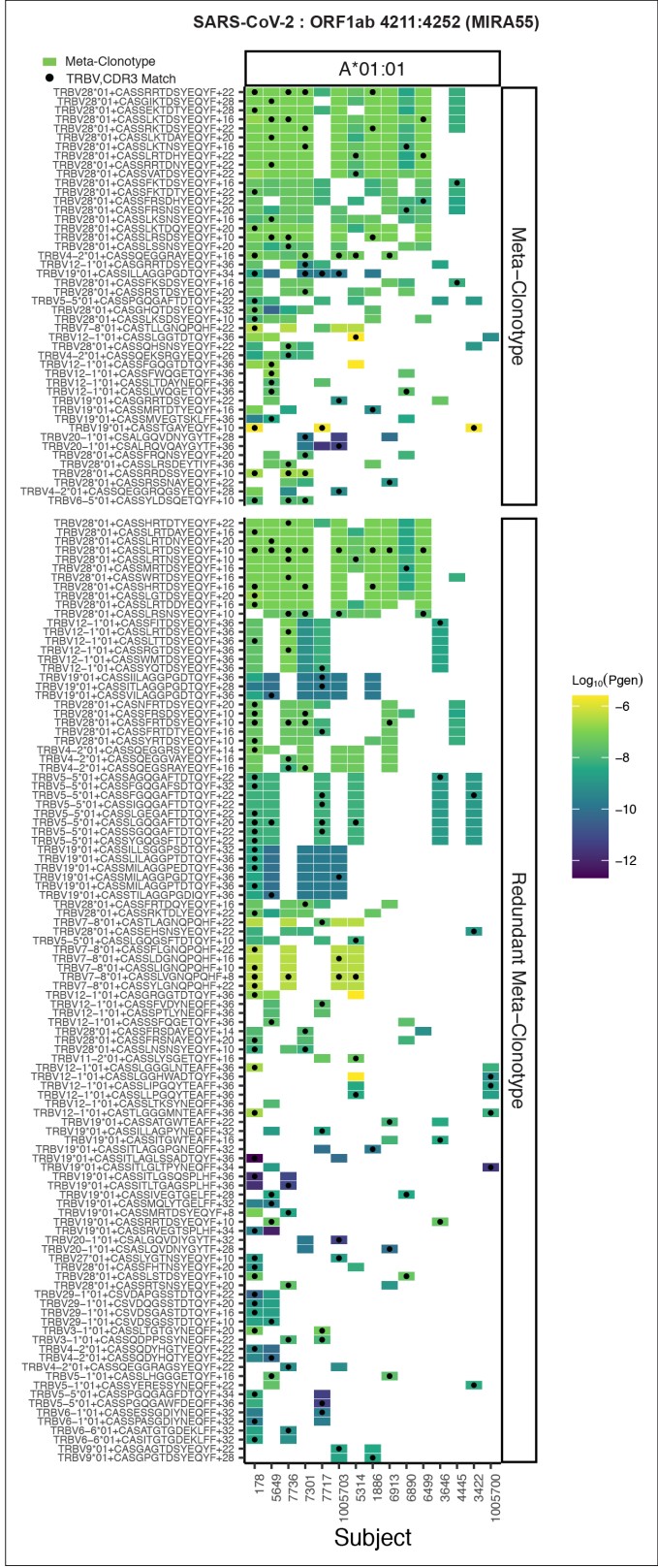

**Figure 6.** Publicity analysis in MIRA participants of CD8+ T-cell receptor (TCR) β-chain features activated by SARS-CoV-2 peptide ORF1ab (MIRA55) predicted to bind HLA-A*01. The grid shows all features that were present in two or more MIRA participants. TCR feature publicity across individuals was assessed using two methods: (1) tcrdist3 *meta-clonotypes* (rectangles) – inclusion criteria defined by a centroid TCR and all TCRs within an optimized

*Figure 6 continued on next page*

*Figure 6 continued*

TCRdist radius selected to span <10⁻⁶ TCRs in a bulk-sequenced background repertoire, and (2) exact public clonotypes (circles) are defined by matching TRBV gene usage and identical complementarity determining region (CDR)3 amino acid sequence. Per subject, the color-scale shows the meta-clonotype conformant clone with the highest probability of generation (p$_{gen}$). All TCRs captured by a 'redundant' meta-clonotypes were completely captured by a higher-ranked meta-clonotype. Redundant meta-clonotypes were not subsequently evaluated.

estimation of the optimal radius. There was minimal further reduction in the estimation bias by using the background that mixed cord blood and OLGA-generated TCRs. We concluded that using a mixed background was desirable because it reduced dependency on sampling from TCRs with specific V–J gene rearrangements, which may be limiting for rarer V–J gene rearrangements; also, the mix helped guard against any idiosyncrasies that may exist in purely synthetic or purely cord blood backgrounds. Ultimately, the best choice for a background may depend on the question being asked and the data that is available, with the ideal background considering factors including donor HLA, age, potential antigen exposures, and other factors that may influence the repertoires.

## Application and meta-clonotype evaluation: engineering meta-clonotypes for SARS-CoV-2

The MIRA antigen stimulation assays used to generate the IMMUNEcode 2.0 database (*Nolan et al., 2020*) identified sets of TCR β-chains associated with recognition of a SARS-CoV-2 antigen using CD8+ T cells enriched from PBMC samples from 62 COVID-19 diagnosed patients and 26 COVID-19-negative subjects. Of these, 253 included at least 6 unique TCRs and included TCRs from more than 1 subject, which we refer to as MIRA0 – MIRA252 based on the number of sequences observed, in descending order (*Supplementary file 1b*). From the MIRA sets, all TCR clonotypes (defined by identical TRBV gene, TRBJ gene, and CDR3 at the amino acid level) were initially considered as candidate centroids; only 2.7 % of the clonotypes were found in more than one MIRA participant. For each candidate TCR, a meta-clonotype was engineered by estimating the maximum radius that limited the estimated number of neighboring TCRs in a bulk antigen-naive repertoire to less than 1 in 10⁶, using a synthetic background as described above. For each MIRA set we then ranked the meta-clonotypes by their sensitivity, approximated as the proportion of TCRs in the set that were within the meta-clonotype radius and matched by CDR3 motif (diagrammed in *Figure 1*). Lower-ranked meta-clonotypes were eliminated from further analysis if all included sequences were completely encompassed by a higher-ranked meta-clonotype; while this reduced redundancy, some overlap remained among meta-clonotypes. We further required that meta-clonotypes be public, including sequences from at least two subjects in the MIRA cohort. We found that 97 of the 252 MIRA sets had sufficiently similar TCRs observed in multiple subjects allowing formation of public meta-clonotypes. From 91,122 TCR β-clonotypes across these 97 MIRA sets – targeting antigens in ORF1ab (*n* = 35), S (*n* = 27), N (*n* = 10), M (*n* = 7), ORF3a (*n* = 7), ORF7a (*n* = 4), E (*n* = 2), ORF8 (*n* = 2), ORF6 (*n* = 1), ORF7b (*n* = 1), and ORF10 (*n* = 1) – we engineered 4548 public meta-clonotypes, which spanned 15 % (13,949/91,122) of the original TCR sequences (*Supplementary file 1f*). The proportion of MIRA antigen-associated TCRs spanned by the meta-clonotypes ranged widely from <1% in MIRA25% to 63% in MIRA7, reflecting broad heterogeneity in the diversity of TCRs inferred as activated by each peptide in the assay.

As an example, the MIRA55 ORF1ab set (TCRs associated with stimulation peptides ALRKVPTD-NYITTY or KVPTDNYITTY) included TCR clonotypes from 15 individuals. From the 449 potential centroids, we defined 40 public meta-clonotypes. Among these features, the radii ranged from 10 to 36 tdus (median 22 tdus), and the publicity – the number of unique subjects spanned by the meta-clonotype – ranged from 3 to 12 individuals (median 6). Meta-clonotype summary statistics for other antigen-associated repertoires are provided in Supplemental Materials (*Supplementary file 1f*). The result was a set of nonredundant, public meta-clonotypes (*Supplementary file 1g*) that could be used to search for and quantify similar putative SARS-CoV-2-specific TCRs in bulk repertoires. In addition to the radius-defined meta-clonotypes (RADIUS), we also developed a modified approach that additionally enforced a sequence motif constraint (RADIUS + MOTIF). The constraint further limited sequence divergence in highly conserved positions of the CDR3, requiring that candidate bulk TCRs match specific amino acids found in the meta-clonotype CDR3s to be counted as part of the neighborhood (see *Figure 1* and Methods).

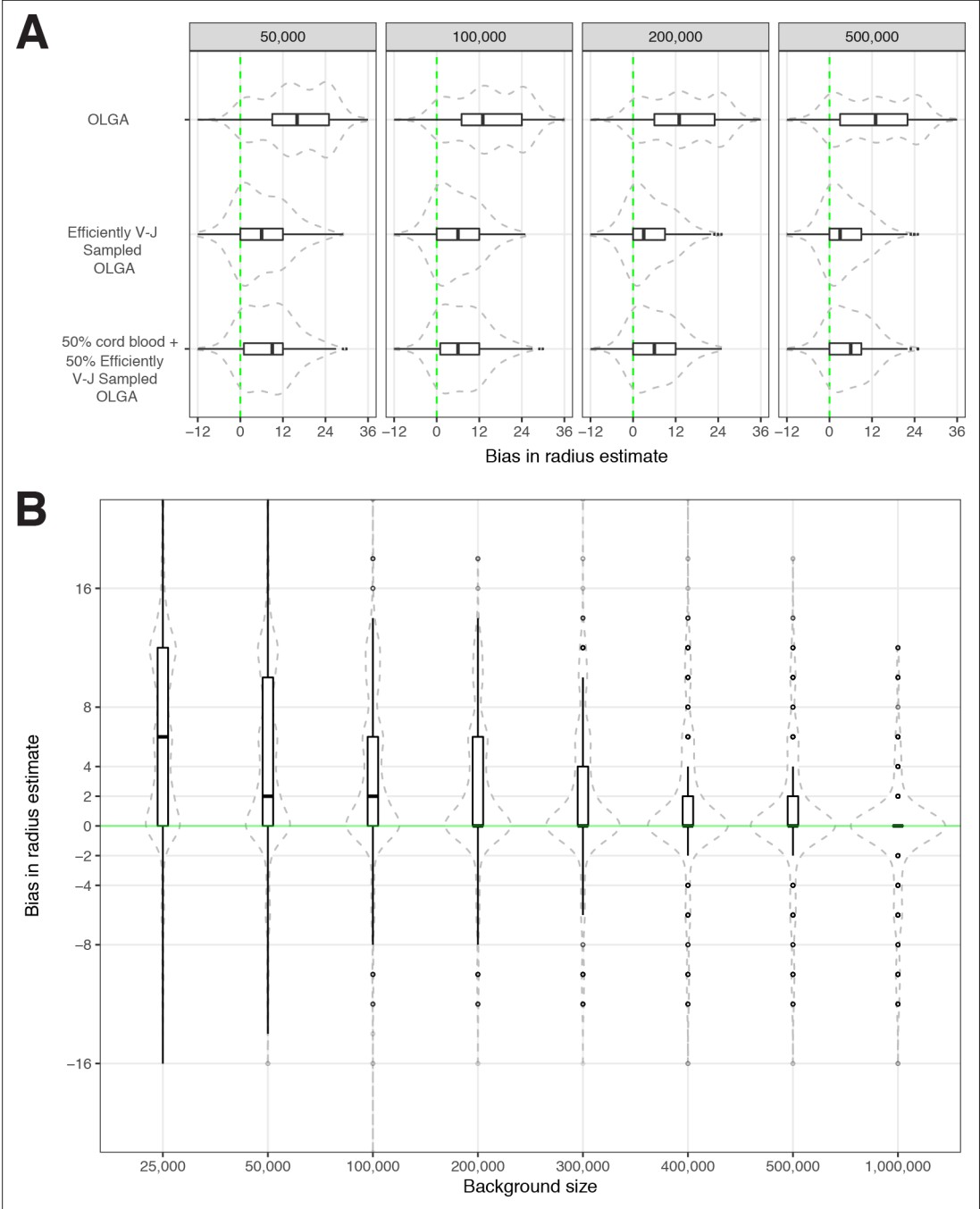

**Figure 7.** Sensitivity of optimized meta-clonotype radius to background size and specification. (**A**) Radius estimates for MIRA55 T-cell receptors (TCRs) using different synthetic backgrounds: (**i**) randomly generated TCRs from Optimized Likelihood estimate of Immunoglobulin Amino acid sequences (OLGA; *Sethna et al., 2019*), (ii) V–J gene-matched sequences generated with OLGA, and (iii) an equal mixture of V–J gene-matched sequences with randomly sampled cord blood TCRs. We compare the estimates generated with the three synthetic backgrounds (of total size 50 , 100 , 200 , and 500 K) to the radii estimates derived using 1 million cord blood TCRs uniformly sampled from eight donors. Weights were applied to correct for biased sampling as described in the paper. (**B**) Evaluation of bias in radius estimates based on background size. Here, we compared bias in subsampled estimate to the estimate derived from a synthetic background of 2 million TCRs (50 % [1 million] cord blood and 50 % [1 million] V–J gene-matched sequences synthesized with OLGA). For each background size, we drew 10 subsamples from the 2 million TCR set.

## Validating meta-clonotypes through confirmation of HLA restriction in COVID-19 patients

Given the integral role of HLA class I molecules in antigen presentation and TCR repertoire selection (*DeWitt et al., 2018*), we further focused on 17 of the 252 MIRA sets that showed strong evidence of HLA-A or HLA-B restriction based on meeting both criteria: (1) computational prediction of HLA binding to the SARS-CoV-2 stimulation peptides, and (2) enrichment of an HLA among participants contributing to the MIRA TCRs. With each set of the MIRA TCRs and the associated SARS-CoV-2 peptides we used HLA-binding predictions (NetMHCpan4.0) to identify the class I HLA alleles that were predicted to bind with strong (IC50 < 50 nM) or weak (50 nm < IC50 < 500 nM) affinity to any of the 8-, 9-, 10-, or 11-mers derived from the stimulation peptides (*Supplementary file 1c and d*). For instance, the peptides associated with MIRA55 TCRs (ORF1ab amino acid positions 1316:1330) are predicted to preferentially bind A*01 (IC50 21 nM), B*15 (IC50 120 nM), and B*35 (IC50 32 nM), and 13 of 13 A*01-positive, and 2 of 34 A*01-negative, patients contributed to the MIRA55 TCR set (Fisher's exact test, p = $10e^{-7}$). We found that for 17 of the MIRA sets, the patients contributing TCRs were significantly enriched for at least 1 HLA-A or HLA-B allele (Fisher's exact test, p < 0.001) (*Supplementary file 1e*). Strong HLA restriction in 17 SARS-CoV-2 MIRA-identified TCR sets provided us an opportunity to validate that meta-clonotypes are antigen-specific features. We hypothesized that in an independent cohort of COVID-19 patients, the abundance of TCRs conforming to each meta-clonotype would be greater in patients having the restricting HLA genotype.

To test this hypothesis, we compared three TCR-based feature sets: (1) radius-defined meta-clonotypes (RADIUS), (2) radius and motif-defined meta-clonotypes (RADIUS + MOTIF), and (3) centroid clonotypes alone, using TRBV-CDR3 amino acid matching (EXACT). Using the features in each set we screened TCRs from the bulk TCR β-chain repertoires of 694 COVID-19 patients whose repertoires were publicly released as part of the immuneRACE datasets (see Methods for details); these patients were not part of the smaller cohort that contributed samples for TCR identification in MIRA experiments. Testing the HLA restriction hypothesis required having the HLA genotype of each individual, which was not provided in the dataset. To overcome this, we inferred each participant's HLA genotype with a classifier that was based on previously published HLA-associated TCR β-chain sequences (*DeWitt et al., 2018*) and their abundance in each patient's repertoire (see Methods for details). MIRA TCRs were not used to assign HLA types to the 694 COVID-19 patients. We then used a beta-binomial counts regression model (*Rytlewski et al., 2019*) with each TCR feature to test for an association of feature abundance with the presence of the restricting allele in the participant's HLA genotype, controlling for participant age, sex, and days since COVID-19 diagnosis. We conducted tests for each meta-clonotype individually, aggregating in each bulk sample the sum of counts of all TCRs conformant with that meta-clonotype's definition. The models revealed that there were radius-defined meta-clonotypes with a strong positive and statistically significant association (FDR <0.01) for 11 of the 17 HLA-restricted MIRA sets that were evaluated (*Figure 8A*; *Supplementary file 1g*); the significant HLA regression odds ratios ranged from 1.4 to 40 (median 4.9), indicating differences in the frequency of meta-clonotype conformant TCRs between patients with and without the HLA genotype. Across all MIRA sets, a positive HLA association (FDR <0.01) was detected for 51.5 % (943/1831) and 59.7 % (1093/1831) of the meta-clonotypes using the RADIUS or RADIUS + MOTIF definitions, respectively. In comparison, an HLA association (FDR <0.01) was detected for fewer than 3.7 % (69/1831) of EXACT centroid features, largely because the specific TRBV gene and CDR3 sequences discovered in the MIRA experiments were infrequently observed in bulk-sequenced samples (*Figure 8B*). When detectable, the abundance of centroid TCRs in bulk repertoires tended to be positively associated with expression of the restricting HLA allele, as hypothesized. However, in most cases, the associated FDR-adjusted *q* value of these associations were orders of magnitude larger (i.e., less statistically significant) than those obtained from using the engineered RADIUS or RADIUS + MOTIF feature with the same clonotype as a centroid (*Figure 9*). The improved performance of meta-clonotypes as query features is particularly evident when testing for HLA-associated enrichment of TCRs recognizing MIRA1 A*01, MIRA48 A*02, MIRA51 A*03, MIRA53 A*24, and MIRA55 A*01 (*Figure 9B*). Moreover, the regression models with meta-clonotypes also revealed possible negative associations between TCR abundance and participant age and positive associations with sample collection more than 2 days post-COVID-19 diagnosis (*Figure 9A*).

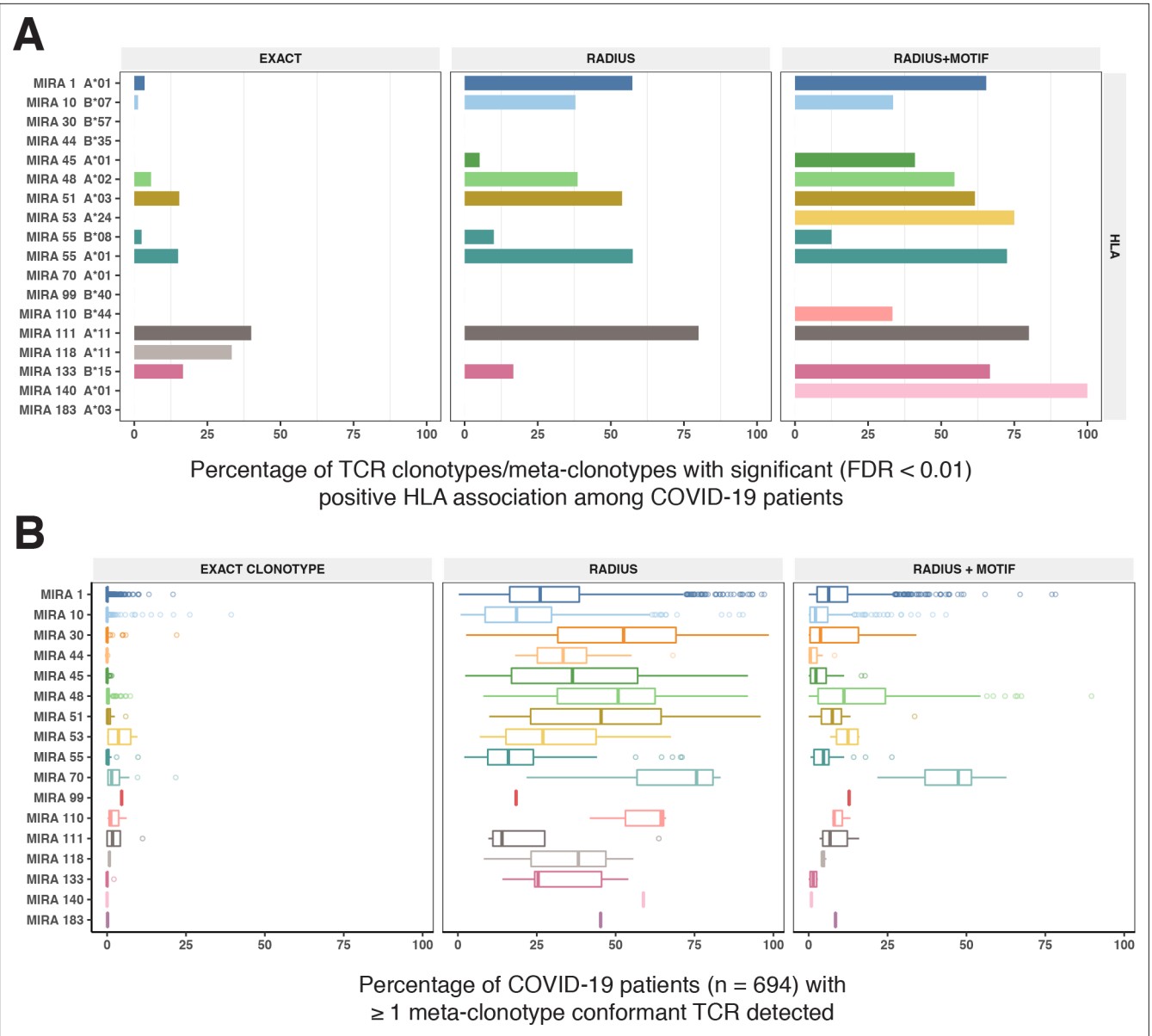

**Figure 8.** HLA restriction of T-cell receptor (TCR) clonotypes and meta-clonotypes in bulk-sequenced TCRβ repertoires of COVID-19 patients. (**A**) Percentage of TCR features with a statistically significant (false discovery rate [FDR] <0.01) association with a restricting HLA allele. We tested for associations between patients' inferred genotype and TCR feature abundance using beta-binomial regression controlling for age, sex, and days since COVID-19 diagnosis. (**B**) For each clonotype/meta-clonotype, the percent of bulk repertoires from COVID-19 patients (*n* = 694) containing TCRs meeting the criteria defined by (1) EXACT (TCRs matching the centroid TRBV gene and amino acid sequence of the complementarity determining region [CDR]3), (2) RADIUS (TCR centroid with inclusion criteria defined by an optimized TCRdist radius), or (3) RADIUS + MOTIF (inclusion criteria defined by TCR centroid, optimized radius, and the CDR3 motif constraint). See *Figure 1* and Methods for details. Meta-clonotype radii were engineered using synthesized backgrounds developed for each MIRA set. Each background contained 100,000 Optimized Likelihood estimate of Immunoglobulin Amino acid sequences (OLGA)-generated TCRs and 100,000 TCRs subsampled from umbilical cord blood; OLGA-generated TCRs were sampled to match to the V–J gene frequency in each MIRA receptor set (i.e., MIRA1, 10, 30, 44, 45, 48, 51, 53, 55, 70, 99, 110, 111,118, 133, 140, or 183) with weighting to account for the sampling bias (see Methods for details).

## Meta-clonotypes provide opportunities to better understand antigen specificity

Since meta-clonotypes cluster similar antigen-annotated TCRs and allow further clustering of similar TCRs from bulk repertoires, they provide an opportunity to visualize and refine hypotheses about the characteristics of TCRs that confer antigen specificity. After observing that in many instances, the

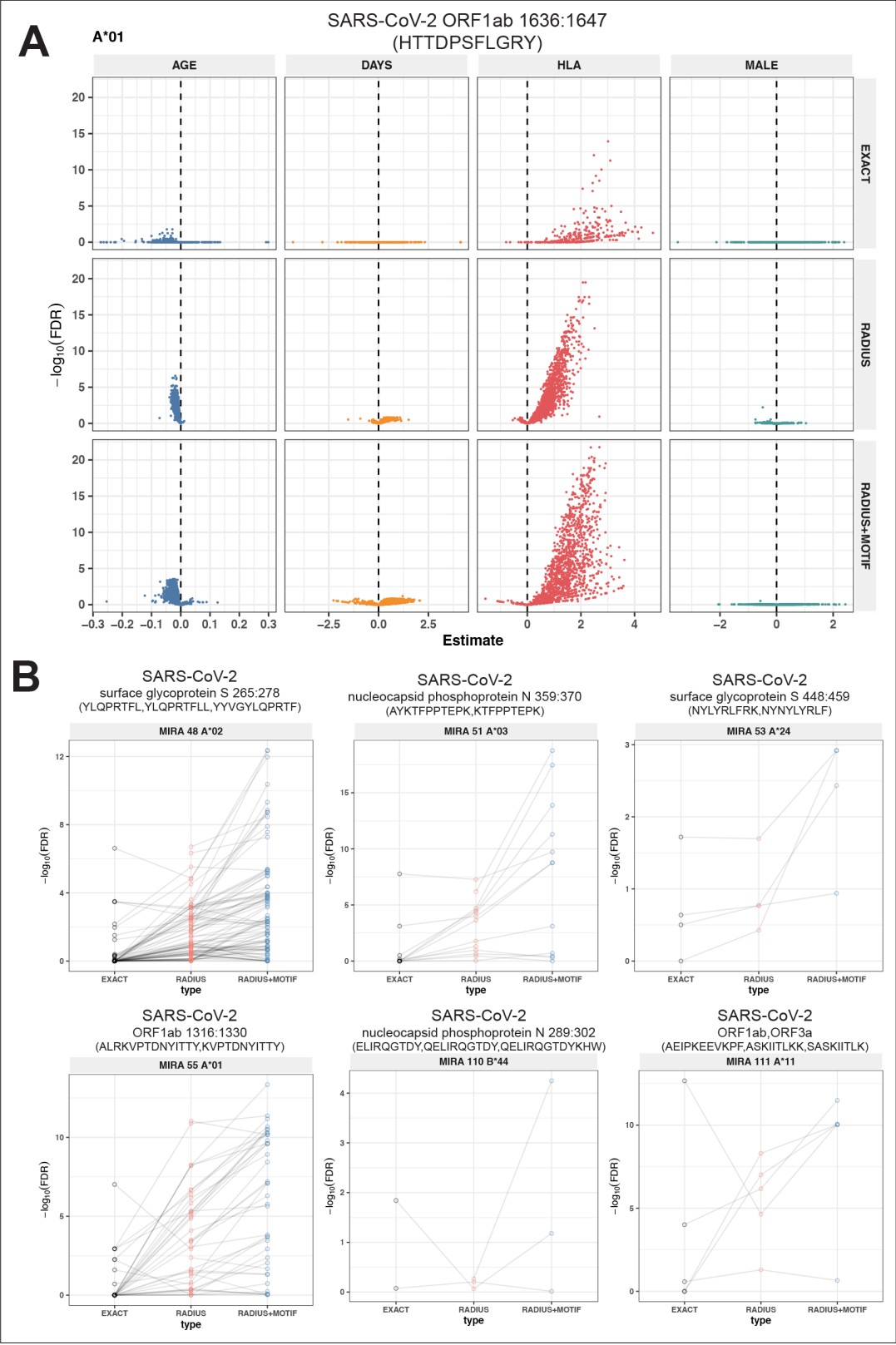

**Figure 9.** Associations of T-cell receptor (TCR) features with participant age, days postdiagnosis, HLA genotype, and sex in TCR β-chain repertoires of COVID-19 patients (*n* = 694). (**A**) Beta-binomial regression coefficient estimates (*x*-axis) and negative log$_{10}$ false discovery rates (*y*-axis) for features developed from CD8+ TCRs activated by SARS-CoV-2 MIRA55 ORF1ab amino acids 1636:1647, HTTDPSFLGRY. The abundances of meta-clonotype

*Figure 9 continued on next page*

*Figure 9 continued*

conformant TCRs are more robustly associated with predicted HLA type than for exact clonotypes. (**B**) Signal strength indicating a positive association between the HLA genotype (two-digit) with TCR β-chain clonotypes (EXACT) and meta-clonotype conformant TCRs (RADIUS or RADIUS + MOTIF), where the restricting HLA genotype was inferred from independent data: (i) MIRA48, (ii) MIRA51, (iii) MIRA53, (iv) MIRA55, (v) MIRA110, and (vi) MIRA111 (*Supplementary file 1f*). Each set of three symbols connected by a line represents an evaluation TCRs conformant to an individual clonotype or a meta-clonotype. Models were estimated with counts of productive TCRs matching a clonotype (EXACT) or conforming to a meta-clonotype (RADIUS or RADIUS + MOTIF) with the following definitions: (1) EXACT (inclusion of TCRs matching the centroid TRBV gene and amino acid sequence of the complementarity determining region [CDR]3), (2) RADIUS (inclusion criteria defined by a TCR centroid and optimized TCRdist radius), and (3) RADIUS + MOTIF (inclusion criteria defined by TCR centroid, optimized radius, and CDR3 motif constraint). See Methods for details. Meta-clonotype radii were engineered using synthesized backgrounds developed for each MIRA set. Each background contained 100,000 Optimized Likelihood estimate of Immunoglobulin Amino acid sequences (OLGA)-generated TCRs and 100,000 TCRs subsampled from umbilical cord blood; OLGA-generated TCRs were sampled to match to the V–J gene frequency in each MIRA receptor set (i.e., MIRA1, 48, 51, 53, 55, 110, or 111) with weighting to account for the sampling bias (see Methods for details).

strength of evidence of HLA-restriction in bulk repertoires was greater for RADIUS + MOTIF versus RADIUS meta-clonotypes, we sought to directly inspect the differences between MOTIF-conformant and non-MOTIF-conformant TCRs and utilize one meta-clonotype as an illustrative example. The MIRA55 meta-clonotype based on the centroid TRBV28*01 + TRBJ27*01 + CASSLKTDAYEQFY provided substantially stronger evidence of HLA association in bulk samples when applied as a RADIUS + MOTIF (radius = 20 tdus, motif = SL[RK][ST][ND].YEQ; FDR-$q$ = 2.4e$^{-10}$) versus as a RADIUS (radius = 20 tdus; FDR-$q$ = 1.4e$^{-5}$) or an individual TCR (i.e., EXACT comparator, FDR-$q$ = 1.0).

To identify critical residues, we constructed a logo plot of all TCR CDR3 amino acid sequences from the COVID-19 patient repertoires that were within the optimal radius (20 tdus) and conformed to the motif constraint (SL[RK][ST][ND].YEQ) together with a 'background-adjusted' logo plot. The background-adjusted plot shows the position-specific Kullback–Leibler divergence from an alignment of background CDR3s that were sampled from cord blood and constrained to use the same V and J genes; it emphasizes the uncommon amino acid residues in the meta-clonotype, reducing the size in particular of residues encoded by the germline V and J genes (*Figure 10A*). Inspection of the logo plots shows that five positions in the CDR3 often contain the amino acids 'LRTDS', which are not commonly found in CDR3s using TRBV28*01 and TRBJ2-7*01. Next, we constructed a second background-adjusted logo from the CDR3s that were within the meta-clonotype radius, but did not conform to the meta-clonotype motif (i.e., RADIUS-ONLY TCRs) (*Figure 10B*). These TCRs are important to characterize because they represented the TCRs that decrease the strength of the meta-clonotype's HLA association and therefore may be more likely to contain nonspecific receptors. RADIUS-ONLY CDR3s tended to differ from the MOTIF conformant sequences at logo positions 6 and 8. The interchangeability of basic amino acids R and K at position 6, which is accommodated by the meta-clonotype MOTIF, was supported by the appearance of both 'LRTDS' and 'LKTDS' in the COVID-19 patient repertoires. However, RADIUS-ONLY TCRs, which may be less likely to recognize the target epitope, frequently differed at position 6, where a G residue or a deletion was present. At position 8, the MOTIF tolerated N or D, as both residues were observed in the aligned MIRA-derived TCRs used to form the meta-clonotype. However, in bulk samples no sequences with N at position eight were detected within 20 tdus of this centroid. An E or a deletion in position eight was common in RADIUS-ONLY neighbors. At position 9, the MOTIF tolerated any amino acid, as there was substantial variability there among the MIRA-derived TCRs, however an S was common in motif-conformant TCRs and A was exclusively found among RADIUS-ONLY TCRs. Together, these analyses, made possible by meta-clonotypes and *tcrdist3*, generate hypotheses about the positions and amino acid residues that are important for antigen specificity.

## Comparison to k-mer-based CDR3 features

Alternative methods exist for generating public TCR features from clustered clonotypes. One strategy is to identify clusters of TCRs that are each uniquely enriched with a short CDR3 k-mer, as implemented in GLIPH2 *Huang et al., 2020*; this approach is well suited for identifying CDR3 k-mers associated with antigenic selection across bulk repertoires when knowledge of the specific antigens is unavailable

**eLife** Research article

Computational and Systems Biology | Immunology and Inflammation

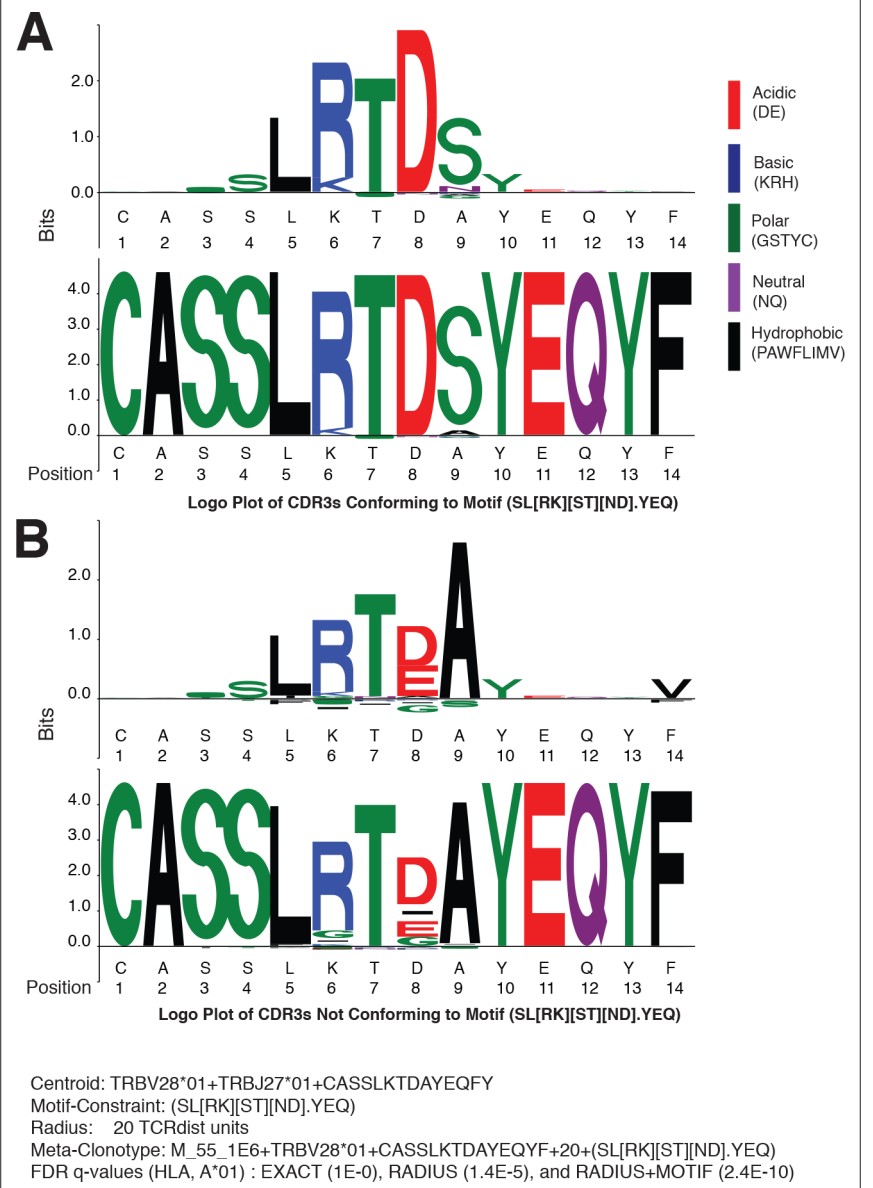

Centroid: TRBV28*01+TRBJ27*01+CASSLKTDAYEQFY
Motif-Constraint: (SL[RK][ST][ND].YEQ)
Radius:    20 TCRdist units
Meta-Clonotype: M_55_1E6+TRBV28*01+CASSLKTDAYEQYF+20+(SL[RK][ST][ND].YEQ)
FDR q-values (HLA, A*01) : EXACT (1E-0), RADIUS (1.4E-5), and RADIUS+MOTIF (2.4E-10)

**Figure 10.** Meta-clonotypes provide opportunities to investigate basis of antigen specificity. Logo plots of T-cell receptors (TCRs) from bulk repertoires of acute and convalescent COVID-19 patients (n = 694) within 20 TCRdist units of MIRA-identified TCR β-chain meta-clonotype M_55_1E6+ TRBV28*01+ CASSLKTDAYEQYF + 20+(SL[RK] [ST][ND].YEQ) centroid. (**A**) Logo plot of TCRs with complementarity determining region (CDR)3 conforming to motif-constraint (SL[RK][ST][ND].YEQ), and (**B**) logo plot of TCRs with CDR3 that do not conform to the motif constraint. The MIRA55 antigen-associated TCR set used to learn the motif included 21 antigen-associated TCRs from 10 subjects. In both panels (**A**) and (**B**), the upper logo motif depicts a 'background-adjusted' logo plot showing the position-specific Kullback–Leibler divergence from an alignment of background CDR3s that were sampled from cord blood TCRs using the same TRBV and TRBJ genes. Lower logo motifs show position-specific amino acid usage. To accommodate CDR3s of different length in the logo plot we aligned each CDR3 to the centroid. The background-adjusted logos are constructed by randomly sampling TCR beta receptors from cord blood with the same TRBV- and TRBJ-gene usage, with 100 V–J-matched TCRs sampled for every receptor in the foreground set.

(**Chiou et al., 2021**). Here, we evaluate the similarities and differences of GLIPH2 and distance-based meta-clonotypes for generating public TCR features from antigen-associated TCRs, by applying both methods to the HLA-restricted MIRA sets (see Methods for details). Both methods identified public molecular patterns from MIRA TCRs (**Figure 11**) that were strongly HLA associated in the large

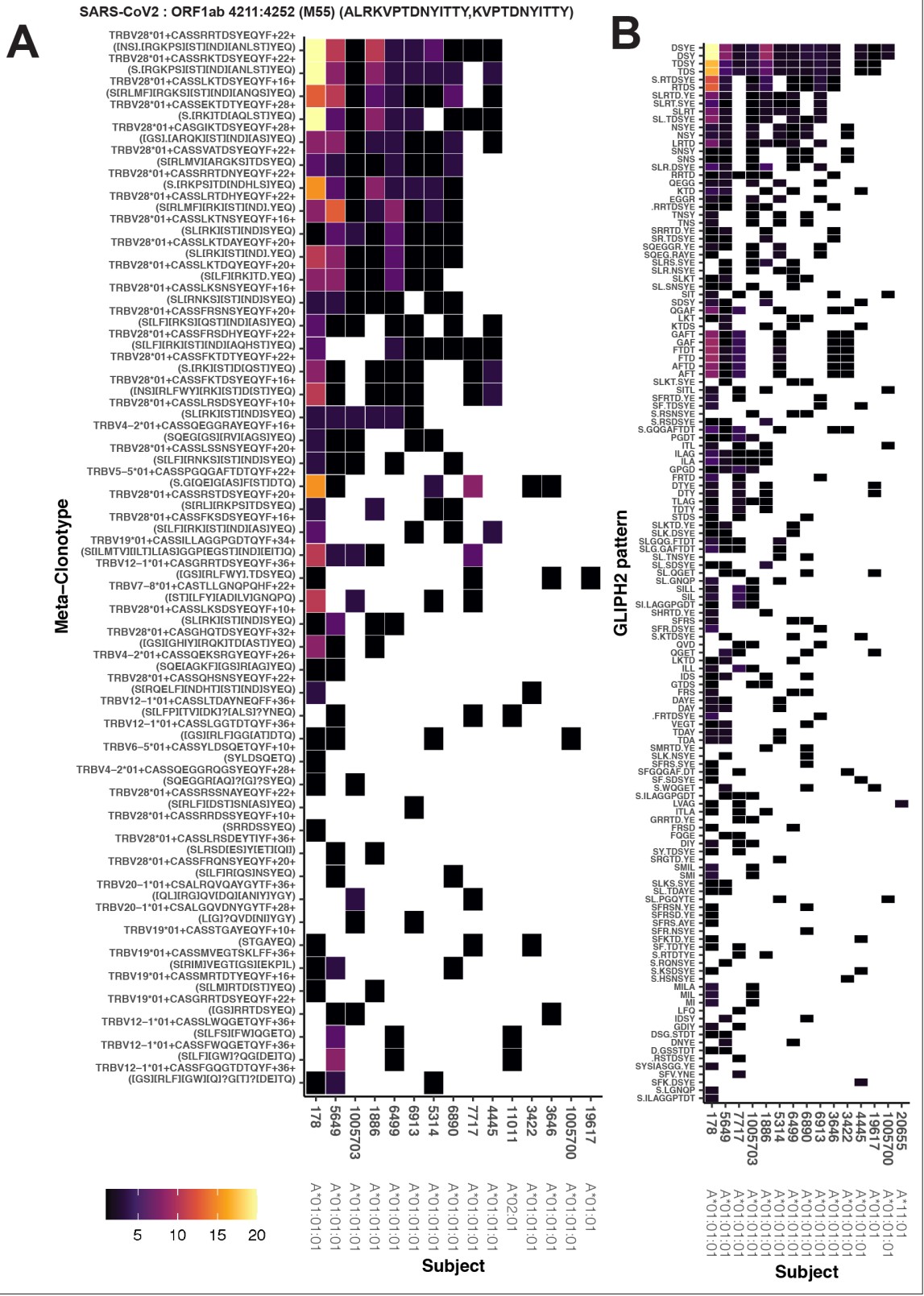

**Figure 11.** Publicity and breadth analysis of CD8+ T-cell receptor (TCR) β-chain features activated by SARS-CoV-2 peptide ORF1ab (MIRA55) using *tcrdist3* and GLIPH2. TCR feature publicity was determined using two methods for clustering similar TCR sequences: (**A**) *tcrdist3*-identified meta-clonotypes and (**B**) GLIPH2 specificity groups, sets of TCRs with a shared complementarity determining region (CDR)3 k-mer pattern uncommon in the program's default background CD8+ receptor data. Grid fill color shows the breadth – or number of conformant clones – within the MIRA-identified clones from each patient.

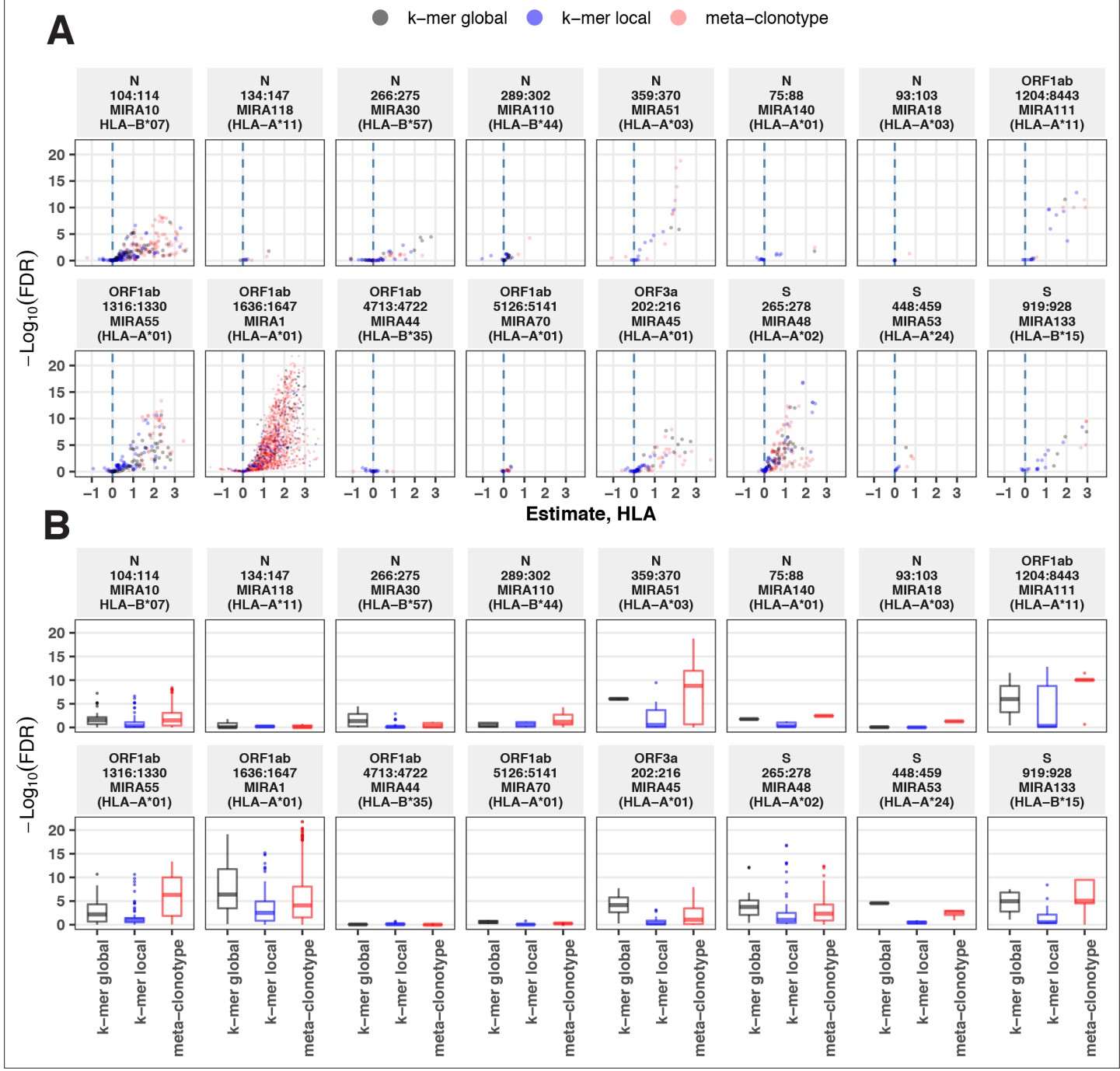

**Figure 12.** Associations between HLA genotypes in COVID-19 patients and abundance of epitope-specific complementarity determining region (CDR)3 k-mers or meta-clonotypes. (**A**) Beta-binomial regression coefficient estimates (x-axis) for participant genotype matching a hypothesized restricting HLA allele and negative $\log_{10}$ false discovery rates (FDRs; y-axis) for features developed from CD8+ T-cell receptors (TCRs) activated by one of 17 HLA-restricted SARS-CoV-2 epitopes found in ORF1ab, ORF3a, nucleocapsid (N), and surface glycoprotein (S). MIRA183 yielded no significant meta-clonotypes (results not shown). Regression models included age, sex, and days postdiagnosis as covariates (not shown). Positive HLA coefficient estimates correspond with greater abundance of the TCR feature in those patients expressing the restricting allele. (**B**) Distribution of FDRs by feature identification method (k-mer local, k-mer global, or meta-clonotype [RADIUS + MOTIF]). Larger negative $\log_{10}$-tranformed FDR values (y-axis) indicate more statistically significant associations. Local k-mer (e.g., FRTD) and global k-mer (e.g., SFRTD.YE) were identified using GLIPH2 (**Huang et al., 2020**) and were used to quantify counts of conforming TCRs in each bulk-sequenced COVID-19 repertoire (see Method for details).

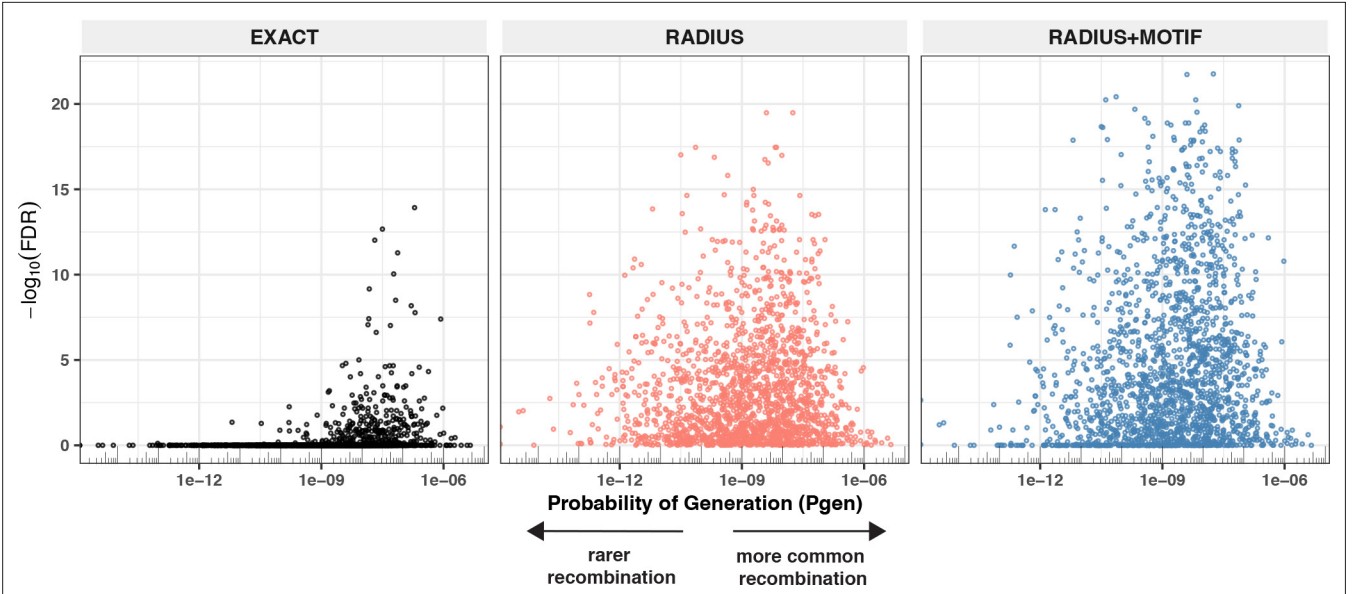

**Figure 13.** Detectable HLA association and complementarity determining region (CDR)3 probability of generation. We evaluated 1831 meta-clonotypes from 17 MIRA sets in a cohort of 694 COVID-19 patients for their association with predicted HLA-restricting alleles. Statistical evidence of the HLA association for each meta-clonotype (RADIUS or RADIUS + MOTIF) and the centroid alone (EXACT) is indicated by the associated false discovery rate (FDR; $y$-axis) in beta-binomial regressions (see Methods for model details). The probability of generation ($p_{gen}$) of each centroid's CDR3-β was estimated using the software OLGA ($x$-axis). Using exact matching, only associations with high probability of generation ($p_{gen}$) antigen-specific T-cell receptors (TCRs) are likely to be detected reliably. However, using meta-clonotypes, *tcrdist3* revealed strong evidence of HLA-restriction for TCRs with both high and low probability of generation. Meta-clonotype radii were engineered using synthesized backgrounds developed for each MIRA set. Each background contained 100,000 Optimized Likelihood estimate of Immunoglobulin Amino acid sequences (OLGA)-generated TCRs and 100,000 TCRs subsampled from umbilical cord blood; OLGA-generated TCRs were sampled to match to the V–J gene frequency in each MIRA receptor set with weighting to account for the sampling bias (see Methods for details).

independent cohort of COVID-19 diagnosed patients (*Figure 12*). For this nonstandard application of GLIPH2, we found that specificity groups based on global CDR3 k-mers (e.g., 'SFRTD.YE') tended to be more consistently HLA associated than specificity groups based on shorter local k-mers (e.g., 'FRTD'). Compared to the GLIPH2 specificity groups based on global CDR3 kmers, meta-clonotypes tended to show similar or more evidence of HLA association (i.e., smaller FDR-$q$ values) (*Figure 12*). MIRA55:ORF1ab is an illustrative example; both the *tcrdist3* meta-clonotypes and GLIPH2-identified TCR groups were more strongly associated with the predicted A*01 HLA restriction than exact clonotypes, supporting the general applicability of using antigen-associated TCRs to create public features from otherwise private antigen-recognizing TCRs. Inspection of the meta-clonotypes and GLIPH2 groups showed that they were often overlapping, with meta-clonotypes subsuming multiple GLIPH2 groups. For example, the A*01-associated meta-clonotype motif S.G[QE]G[AS]F[ST]DTQ (p value 1E−12) fully overlaps several A*01-associated GLIPH2 patterns including S.GQGAFTDT (p value 1E−12), QGAF (p value 1E-11), and SLG.GAFTDT (p value 1E−6). Similarly, the A*01-associated meta-clonotype motif S[RLMF][RK][ST][ND].YEQ (p value 1E−13) covers 21 global GLIPH motifs including SFRTD.YE (p value 1E−10), SLRTD.YE (p value 1E−7), and SF.TDSYE (p value 1E−4) (*Supplementary file 1i*). These observations suggest that the motif constraints of the meta-clonotypes were able to match a broader set of antigen-specific CDR3s compared to any one GLIPH2 specificity pattern, which may have helped boost detection sensitivity in the COVID-19 repertoires.

## Discussion

Given the extent of TCR diversity, only antigen-associated TCRs with high probability of generation ($p_{gen}$) are likely to be detected reliably across individuals (*Figure 13*). While public, high-$p_{gen}$ TCRs may sometimes be available for detecting a prior antigen exposure, to better understand the population-level dynamics of complex polyclonal T-cell responses across a gradient of generation probabilities, it

is critical to develop methods for finding public meta-clonotypes that capture otherwise private TCRs (*Figure 13*). We developed a novel framework, leveraging antigen-associated TCRs and efficiently sampled background repertoires, to engineer meta-clonotypes that balance the need for sufficiently public features with the need to maintain antigen specificity. The output of the analysis framework (*Figure 1*) is a set of portable meta-clonotypes, each defined by a (1) centroid, (2) radius, and (3) a CDR3 motif pattern, that can be used to rapidly search bulk repertoires for similar TCRs that likely share a cognate antigen. To demonstrate this analytical framework, we analyzed publicly available sets of antigen-associated TCR β-chain sequences that putatively recognize SARS-CoV-2 peptides (*Nolan et al., 2020*). From these, we generated 4548 TCR radius-defined public meta-clonotypes that can be used to further investigate CD8+ T- cell response to SARS-CoV-2 (*Supplementary file 1G and H*).

To evaluate the properties of radius-defined meta-clonotypes we utilized the immuneRACE dataset and focused on the SARS-CoV-2 epitopes with the strongest evidence of HLA restriction (*Supplementary file 1g*, $n$ = 1831 associated meta-clonotypes). We reasoned that we could compare the abundance of meta-clonotypes in COVID-19 patients with and without the restricting HLA genotype, and that a significant positive association of abundance with the restricting genotype would provide confirmatory evidence of the meta-clonotype's SARS-CoV-2 antigen specificity in addition to its HLA restriction. Overall, we found significant confirmatory evidence of the HLA restriction of meta-clonotype abundance for a majority of the MIRA sets we analyzed (11/17, 64%) and for a majority (1080/1831, 59%) of the individual meta-clonotypes tested using the RADIUS + MOTIF approach; importantly, there were no meta-clonotypes significantly associated with the absence of expression of the restricting HLA allele. There are several plausible explanations for the remaining meta-clonotypes that did not have a significant signal of HLA restriction in this study. One possibility is that meta-clonotype definitions were not sufficiently specific for the target antigen; the radius is optimized for specificity, but not all amino acid substitutions accommodated within the radius are guaranteed to preserve antigen recognition, and while the motif constraint increases specificity, it is likely that meta-clonotype definitions could be further refined with more antigen-associated TCR data and enhanced motif refinement methods. Also, subdominant SARS-CoV-2 epitopes may not be ubiquitously presented, even among participants that share the required HLA genotype, which weakens the signal of HLA restriction detectable by regression analysis.

The meta-clonotype framework we present joins a class of commonly used methods for TCR analysis that depend on comparisons to an antigen-naive background repertoire. For example, GLIPH2 attempts to find CDR3 $k$-mers that are significantly more frequent in the data compared to a naive background and TCRNET organizes repertoires into networks to identify nodes with an enriched number of edges compared to the number of edges formed within a background repertoire. An important distinction is that the meta-clonotype framework is designed to leverage data that has been experimentally pre-enriched with antigen-specific TCRs. In contrast, GLIPH2 and TCRNET have been designed to identify clusters of similar TCRs in a bulk repertoire and calibrated to find statistically significant enrichment of each cluster in the bulk repertoire compared to a background. This distinction is important because among a set of TCRs enriched for antigen specificity, it is possible that many CDR3 signatures (or network nodes) might be statistically enriched compared to a background, yet they may still be abundant in the background and lack sufficient specificity for subsequent analyses of bulk repertoires. We saw evidence of this in our comparison with GLIPH2; while many 'global' GLIPH2 groups performed similar to meta-clonotypes, the short 'local' groups identified by the algorithm were often too short to be specific, and they were generally not as strongly associated with the restricting HLA in our analysis. The meta-clonotype approach is distinct because it estimates an optimal radius to control the probability of finding conformant TCRs in a naive background at a prespecified level (i.e., 1 in 1 million). This probability is further reduced by the CDR3 motif constraint, which requires that conformant TCRs match at least one of the residues in the antigen-associated sequences at critical conserved positions; other methods developed primarily for identifying TCRs under antigenic selection from bulk repertoire data do not leverage this information. Finding a radius to control the frequency of meta-clonotype conformant TCRs in the background was also made more efficient by a background sampling algorithm that focused on TCRs with matching V and J genes with weighting to account for the sampling bias. Ultimately, it is the focus on controlling the absolute frequency of meta-clonotype conformant TCRs in an antigen-naive background that gives the meta-clonotype definitions portability to be

applied to analyses of bulk repertoires, where quantification of similar antigen-specific TCRs is required.

Recently, *Snyder et al., 2020* analyzed 1521 bulk TCR β-chain repertoires from COVID-19 patients in the immuneRACE dataset and an additional 3500 (not yet publicly available) repertoires from healthy controls to identify public TCR β-chains that could be used to identify SARS-COV-2 infected individuals with high sensitivity and specificity. Their results show that with sufficient data it is possible to engineer performant TCR biomarkers of antigen exposure from exact clonotypes. We show that by leveraging antigen-associated TCR repertoires it is possible to engineer meta-clonotypes from a relatively small group of COVID-19 diagnosed individuals ($n$ = 62; HLA-typed $n$ = 47), with TCRs conformant to these meta-clonotypes frequently detectable in a larger independent cohort. We propose that meta-clonotypes constitute a set of potential features that could be leveraged in developing TCR-based clinical biomarkers that go beyond detection of infection or exposure. For example, biomarkers predictive of infection, disease severity, or vaccine protection may each require different TCR features. We note that meta-clonotypes are often overlapping in that a single TCR may be conformant with the definition of multiple meta-clonotypes, which should be a consideration when applying a set of meta-clonotypes together. For instance, tallying conformant TCRs in a repertoire should avoid counting a TCR more than once, while in a biomarker context many statistical and machine learning algorithms may benefit from a set of partially redundant features to amplify the clinically relevant signals. We also note that it may be more immunologically relevant to quantitate the frequency of *unique* clonotypes conforming to a meta-clonotype in each repertoire (i.e., clonal breadth) in addition to the overall frequency; it is plausible both may carry important signals. Much like any biomarker study, to establish a TCR-based predictor of a particular outcome, the features must be measured among a sufficiently large cohort of individuals, with a sufficient mix of outcomes; meta-clonotypes offer a way to build public features that are suitable for this process.

Though demonstrating HLA restriction of the SARS-CoV-2 meta-clonotypes establishes their potential utility, it also highlighted how HLA diversity could be a major hurdle to biomarker development. The sensitivity of a TCR-based biomarker in a diverse population may depend on combining meta-clonotypes with diverse HLA restrictions since individuals with different HLA genotypes often target different epitopes using divergent TCRs. Our analysis shows that having HLA genotype information for TCR repertoire analysis can be critical to interpreting results. The simple HLA classifier we developed suggests that soon it may be possible to infer high-resolution HLA genotype from bulk TCR repertoires, but until then it is valuable to have sequenced-based HLA genotyping. In the absence of HLA genotype information, it may still be feasible to generate informative TCR meta-clonotypes. For example, a poly-antigenic TCR-enrichment strategy (i.e., peptide pools or whole proteins) could help generate meta-clonotypes that broadly cover HLA diversity if the sample donors are racially, ethnically, and geographically representative of the ultimate target population. For these reasons, donor unrestricted T cells and their receptors (e.g., MAITs and γδ T cells) may also be good targets for TCR biomarker development.

To enable TCR biomarker development and innovative extensions of distance-based immune repertoire analysis, we developed *tcrdist3*, which provides open-source (https://github.com/kmayerb/tcrdist3), documented (https://tcrdist3.readthedocs.io) computational building blocks for a wide array of TCR repertoire workflows in Python3. The software is highly flexible, allowing for: (1) customization of the distance metric with position and CDR-specific weights and amino acid substitution matrices, (2) inclusion of CDRs beyond the CDR3, (3) clustering based on single-chain or paired-chain data for α/β or γ/δ TCRs, and (4) use of default as well as user-provided TCR repertoires as background for controlling meta-clonotype specificity (e.g., users may want to use HLA genotype-matched, or age-matched backgrounds). *tcrdist3* makes efficient use of available CPU and memory resources; as a reference, identification of meta-clonotypes from the MIRA55:ORF1ab dataset ($n$ = 479 TCRs) was completed in less than 5 min using 2 CPUs and <4 GB of memory including distance computation and radius optimization. Quantification of the identified meta-clonotypes ($n$ = 40) conformant TCRs in 694 bulk β-chain repertoires, ranging in size from 10,395 to 1,038,012 in-frame clones (~5 billion total pairwise comparisons) could be completed in less than 2 hr using 2 CPUs and <6 GB memory. The package also can generate multiple types of publication-ready figures (e.g., background-adjusted CDR3 sequence logos, paired TRAV–TRAJ/TRBV–TRBJ-gene usage chord diagrams, and annotated TCR dendrograms). The continued maturation of multiple adaptive immune receptor repertoire

sequencing technologies will open possibilities for basic immunology and clinical applications, and *tcrdist3* provides a flexible tool that researchers can use to integrate the data sources needed to detect and quantify antigen-specific TCR features.

# Materials and methods

**Key resources table**

| Reagent type (species) or resource | Designation | Source or reference | Identifiers | Additional information |
|---|---|---|---|---|
| Software, algorithm | Python3, Numpy, Pandas, | | Python Programming Language, RRID:SCR_008394 NumPy, RRID:SCR_00863 Pandas, RRID:SCR_01821 | |
| Software, algorithm | R, ggplot2 | | R Project for Statistical Computing, RRID:SCR_001905 ggplot2, RRID:SCR_014601 | |
| Software, algorithm | tcrdist3 | This study | tcrdist3 0.2.0 | https://github.com/kmayerb/tcrdist3 |
| Software, algorithm | pwseqdist | This study | pwseqdist 0.5 | https://github.com/agartland/pwseqdist |
| Script, algorithm | hla3 | This study | version 0.1.0 | https://github.com/kmayerb/hla3 |
| Software, algorithm | corncob | *Martin et al., 2020* doi:10.1214/19-aoas1283 | | https://github.com/bryandmartin/corncob (*Martin, 2021*) |
| Software, algorithm | OLGA | *Sethna et al., 2019* 10.1093/bioinformatics/btz035 | | https://github.com/statbiophys/OLGA (*Isacchini, 2021*) See slight modifications in: https://github.com/kmayerb/tcrdist3/blob/master/tcrdist/olga_directed.py |
| Software, algorithm | GLIPH2 | *Huang et al., 2020* 10.1038/s41587-020-0505-4 | version 2 | http://50.255.35.37:8,080 |

## TCR data: immuneRACE datasets and MIRA assay

The study utilized two primary sources of TCR data (*Nolan et al., 2020*; *Snyder et al., 2020*). The first data source was a table of TCR β-chains amplified from CD8+ T cells activated after exposure to a pool of SARS-CoV-2 peptides, using a Multiplex Identification of Receptor Antigen (MIRA) (*Klinger et al., 2015*); data were accessed July 21, 2020 and labeled 'ImmuneCODE-MIRA-Release002'. The samples used for the MIRA analysis included samples from 62 individuals diagnosed (3 acute, 1 nonacute, 58 convalescent) with COVID-19, of whom 47 (3 acute, 44 convalescent) were HLA genotyped in the ImmuneCODE-MIRA-Release002 *subject-metadata.csv* file. When assessing the frequency of neighboring TCRs in antigen-associated MIRA sets, we also used TCRs evaluated by MIRA from 26 COVID-19-negative control subjects activate by SARS-CoV-2 peptides that were part of ImmuneCODE-MIRA-Release002. We analyzed the 253 MIRA sets with at least six unique TCRs contributed by ≥2 people, referred to as MIRA0-MIRA252 in rank order by their size (*Supplementary file 1b*); each 'MIRA set' included antigen-associated TCRs across all assayed individuals. Adaptive Biotechnologies also made publicly available bulk-sequenced TCR β-chain repertoires from COVID-19 patients participating in a collaborative immuneRACE network of international clinical trials. We analyzed repertoires from 694 individuals where meta-data were available indicating that the sample was collected from 0 to 30 days from the time of diagnosis. COVID-19-DLS (Alabama, USA, *n* = 374); COVID-19-HUniv12Oct (Madrid, Spain, *n* = 117); COVID-19-NIH/NIAID (Pavia, Italy, *n* = 125)+ COVID-19-ISB (Washington, USA, *n* = 78). The sampling depth of these repertoires varied from 15,626 to 1,220,991 productive templates (median 208,709) and 10,395–1,038,012 productive rearrangements (median 113,716). We did not use bulk samples from the COVID-19-ADAPTIVE dataset as the average age was substantially lower than other immuneRACE populations and to avoid possible overlap with individuals that contributed samples to the MIRA experiments.

## HLA genotype inferences

No publicly available HLA genotyping was available for the 694 bulk-sequenced immuneRACE T-cell repertoires (*Nolan et al., 2020*). Before considering SARS-CoV-2-specific features, we inferred the HLA genotypes of these participants based on their TCR repertoires. Predictions were based on previously published HLA-associated TCR β-chain sequences (*DeWitt et al., 2018*) and their detection in each repertoire. Briefly, a weight-of-evidence classifier for each HLA loci was computed as follows: For each sample and for each common allele, the number of detected HLA-diagnostic TCR β-chains was divided by the total possible number of HLA-diagnostic TCR β-chains. The weights were normalized as a probability vector and the two highest HLA-allele probabilities (if the probability was larger than 0.1) were assigned to each repertoire; homozygosity was inferred if only one allele had probability >0.1. The sensitivity and specificity of this simple classifier for each allele prediction were assessed using 550 HLA-typed bulk repertoires (*Emerson et al., 2017*). Sensitivities for common alleles A*01:01, A*02:01, A*03:01, A*24:02, A*11:01, B*07:02, B*44:02, B*15:01, B*35:01, B*40:01, and B*57:01 were between 0.85 and 1. Specificities for these major HLA-A and HLA-B alleles were between 0.97 and 1.0. Inference of the HLA genotype of most participants was deemed sufficient in the absence of direct HLA genotyping. This weight-of-evidence predictor is implemented as an open-source python script (https://github.com/kmayerb/hla3, copy archived at swh:1:rev:daaa03b89883629e53974c8e-5cab2563971acfa0, *Mayer-Blackwell, 2021a*).

## Peptide–HLA-binding prediction

HLA-binding affinities of peptides used in the MIRA stimulation assay were computationally predicted using NetMHCpan4.0 (*Jurtz et al., 2017*). Specifically, the affinities of all 8-, 9-, 10-, and 11-mer peptides derived from the stimulation peptides were computed with each of the class I HLA alleles expressed by participants in the MIRA cohort (*n* = 47). From these data, we derived two-digit HLA-binding predictions (e.g., A*02) for each MIRA set by pooling the predictions for all the four-digit HLA variants (e.g., A*02:01, A*02:02) across all the derivative peptides and selecting the lowest IC50 (strongest affinity). Predictions with IC50 <50 nM were considered strong binders and IC50 <500 nM were considered weak binders (*Supplementary file 1c and d*).

## TCR distances

Weighted multi-CDR distances between TCRs were computed using *tcrdist3*, an open-source Python3 package for TCR repertoire analysis and visualization, using the procedure first described in *Dash et al., 2017*. The package has been expanded to accommodate γδ TCRs; it has also been recoded to increase CPU efficiency using *numba*, a high-performance just-in-time compiler. A numba-coded edit/Levenshtein distance is also included for comparison.

Briefly, the distance metric in this study is based on comparing TCR β-chain sequences. The *tcrdist3* default settings compare TCRs at the CDR1, CDR2, and CDR2.5 and CDR3 positions. By default, IMGT aligned CDR1, CDR2, and CDR2.5 amino acids are inferred from TRVB gene names, using the *01 allele sequences when allele-level information is not available. The CDR3 junction sequences are trimmed three amino acids on the N-terminal side and two amino acids on the C-terminus, positions that are highly conserved and less crucial for mediation of antigen recognition. For two CDR3s with different lengths, a set of consecutive gaps are inserted at a position in the shorter sequence that minimizes the summed substitution penalties based on a BLOSUM62 substitution matrix. Insertions are penalized as nonconservative amino acid substitutions. Distances are then the weighted sum of substitution penalties across all CDRs, with the CDR3 penalty weighted three times that of the other CDRs.

## Synthetic TCR backgrounds

To estimate optimal radii at which background TCRs are expected to be detected at a frequency of $<10^{-6}$, for each antigen-associated MIRA set, we constructed synthetic backgrounds that combine efficient sampling of OLGA-generated TCR sequences V–J matched to the MIRA set with randomly sampled antigen-naive TCRs from eight cord blood donors (*Britanova et al., 2016*). A slightly modified version of OLGA (Optimized Likelihood estimate of Immunoglobulin Amino acid sequences) from *Sethna et al., 2019* was used to efficiently generate V–J gene-matched CDRs based on a previously trained statistical model of VDJ recombination (*Marcou et al., 2018*). Prevalence weights were

assigned to the OLGA sequences to correct for the oversampling of specific V–J gene pairings in the synthetic background. The expected prevalence of TCRs using a given V–J gene pairing was inferred from natural frequencies observed in the cord blood data. The slightly modified version of OLGA source code used to generate synthetic TCRs directed by selected V and J genes is contained within the *tcrdist3* source code.

## TCR meta-clonotype MOTIF constraint

Radius-optimized meta-clonotypes from antigen-associated TCRs provided an opportunity to discover key conserved residues most likely mediating antigen specificity. We developed a 'motif' constraint as an optional part of each meta-clonotype definition that limited allowable amino acid substitutions in highly conserved positions of the CDR3 to those observed in the antigen-associated TCRs. The motif constraint for each radius-defined meta-clonotype was defined by aligning each of the conformant CDR3 amino acid sequences to the centroid CDR3. Alignment positions with five or fewer distinct amino acids were considered conserved and added to the motif as a set of possible residues. Thus, the motif constraint is permissive of only specific substitutions in select positions relative to the centroid, however these substitutions are still penalized by the radius constraint. The motif constraint was encoded as a regular expression, with the '.' character indicating nonconserved positions and bracketed residues indicating a degenerate position with a set of allowable residues (e.g., 'SL[RK][ND]YEQ'). Position with gaps, where some sequences are missing a residue, are accommodated by making that position optional (e.g., 'SL[RK]?[ND]YEQ'). Since the motif constraints form regular expressions, they can be used to rapidly scan large repertoires for conformant TCRs and easily be combined with a radius constraint. When applied to bulk repertoires, the motif constraint eliminates CDR3s that did not match key conserved residues.

## Quantifying meta-clonotype conformant TCRs in bulk repertoires

After defining a set of meta-clonotypes using antigen-associated TCRs, we searched for similar TCRs in 694 bulk repertoires from COVID-19 patients 0–30 days from diagnosis. Association with predicted HLA was tested based on the count TCRs conformant with each meta-clonotype individually; however, we note that meta-clonotypes are often overlapping in that a single TCR may be conformant with the definition of multiple meta-clonotypes. A full example of tabulating EXACT, RADIUS, and RADUS+ MOTIF meta-clonotypes conformant TCRs in a bulk repertoire, while avoiding such double counting, is provided in *tcrdist3* documentation page (https://tcrdist3.readthedocs.io).

## Abundance regression modeling

Similar to bulk RNA sequencing data, TCR frequencies are count data drawn from samples of heterogeneous size. Thus, we initially attempted to fit a negative binomial model to the data, for example, DESEQ2 (*Love et al., 2013*). We found that the negative binomial model did not adequately fit TCR counts, which – compared to transcriptomic data – were characterized by (1) more technical zeros due to inevitable under sampling and (2) even greater biological overdispersion, which could be due to clonal expansions and HLA genotype diversity. Instead we found that the beta-binomial distribution, which was recently used for TCR abundance modeling (*Rytlewski et al., 2019*), provided the flexibility needed to adequately fit the TCR data. We used an R package, *corncob*, which provides maximum likelihood methods for inference and hypothesis testing with beta-binomial regression models (*Martin et al., 2020*). Due to the sparsity of some meta-clonotypes, 7% of coefficient estimates in regression models had p values larger than 0.99 (i.e., nonsignificant) and unreliable high magnitude coefficient estimates. These values are not shown in the horizontal range of the volcano plots. From the p values for each regression coefficient we computed FDR-adjusted $q$ values and accepted $q$ values <0.01 (1%) as statistically significant; adjustment was performed across meta-clonotypes within each MIRA set and within each variable class (e.g., HLA, age, sex, or days since diagnosis). The HLA regression coefficients from the beta-binomial models indicate log-fold differences in meta-clonotype abundance between patients with and without the HLA genotype.

## Comparison with k-mer-based CDR3 features

*GLIPH2* (*Huang et al., 2020*) software *irtools.osx* was applied to 17 antigen-associated subrepertoire of TCRs with epitopes with strong prior evidence of restriction to an HLA-A or HLA-B allele to

demonstrate how a k-mer-based tool might also be used to cluster biochemically similar antigen-specific TCRs to discover potential TCR biomarker features. GLIPH2 generates 'global' TCR specificity groups of CDR3s of identical length with a single optional nonconserved position based on enrichment frequency of 'local' continuous 2-, 3-, and 4-mers. We used the GLIPH2-provided 'ref_CD8_v2.0.txt' background file as a background to identify enriched features. Across epitope-specific MIRA sets, we tested HLA associations of 812 GLIPH2 pattern ranging from 3 to 11 amino acids in length. The MIRA55:ORF1ab set was chosen for detailed analysis because, among the MIRA sets, it is comprised of CD8+ TCR β-chains activated by a peptide with the strongest evidence of HLA restriction, primarily HLA-A*01. The MIRA55 set of TCRs, GLIPH2 returned 121 testable public clusters based on 67 local k-mers (e.g., FRTD) and 54 global k-mer (e.g., SFRTD.YE), associated with CDR3 patterns enriched relative the program's default CD8+ TCR background (GLIPH2 default Fisher's exact test, p value <0.001). The GLIPH2 patterns and their associated 'specificity group' TRBV gene usages and sequence length were then used to search for conforming TCRs in the 694 bulk-sequenced COVID-19 repertoires, allowing comparison to exact and meta-clonotype features. GLIPH2 represents degenerate positions using the '%' character, which we represent throughout this study by the '.' character.

## Tcrdist3: software for TCR repertoire analysis

*tcrdist3* is an open-source Python3 package for TCR repertoire analysis and visualization. The core of the package is the TCRdist, a distance metric for relating two TCRs, which has been expanded beyond what was previously published (*Dash et al., 2017*) to include γδ-TCRs. It has also been recoded to increase CPU efficiency using *numba*, a high-performance just-in-time compiler. A numba-coded edit/Levenshtein distance is also included for comparison, with the flexibility to accommodate novel TCR metrics as they are developed. The package can accommodate data in standardized format including AIRR, vdjdb exports, MIXCR output, 10× Cell Ranger output or Adaptive Biotechnologies immunoSeq output. The package is well documented including examples and tutorials, with source code available on github.com under an MIT license (https://github.com/kmayerb/tcrdist3; *Mayer-Blackwell, 2021b*). *tcrdist3* imports modules from several other open-source, pip installable packages by the same authors that support the functionality of *tcrdist3*, while also providing more general utility. Briefly, the novel features of these packages and their relevance for TCR repertoire analysis is described here: *pwseqdist* enables fast and flexible computation of pairwise sequence-based distances using either *numba*-enabled tcrdist and edit distances or any user-coded Python3 metric to relate TCRs; it can also accommodate computation of 'rectangular' pairwise matrices: distances between a relatively small set of TCRs with all TCRs in a much larger set (e.g., bulk repertoire). On a modern laptop, distances can be computed at a rate of ~70 M per minute, per CPU.

*tcrsampler* is a tool for subsampling large bulk datasets to estimate the frequency of TCRs and TCR neighborhoods in background repertoires. The module comes with large, bulk sequenced, default databases for human TCR α, β, γ, and δ and mouse TCR β (*Britanova et al., 2016*; *Ravens et al., 2018*; *Wirasinha et al., 2018*). Datasets were selected because they represented the largest preantigen exposure TCR repertoires available; users can optionally supply their own background repertoires when applicable. An important feature of *tcrsampler* is the ability to specify sampling strata; for example, sampling is stratified on individual by default so that results are not biased by one individual with deeper sequencing. Sampling can also be stratified on V- and/or J-gene usage to oversample TCRs that are somewhat similar to the TCR neighborhood of interest. This greatly improves sampling efficiency, since comparing a TCR neighborhood to a background set of completely unrelated TCRs is computationally inefficient; however, we note that it is important to adjust for biased sampling approaches to estimate the frequency of oversampled TCRs in a bulk-sequenced repertoire.

*palmotif* is a collection of functions for computing symbol heights for sequence logo plots and rendering them as SVG graphics for integration with interactive HTML visualizations or print publication. Much of the computation is based on existing methods that use either KL divergence/entropy or odds ratio-based approaches to calculate symbol heights. We contribute a novel method for creating a logo from CDR3s with varying lengths. The target sequences are first globally aligned (parasail C++ implementation of Needleman–Wunsch) to a preselected centroid sequence (*Daily, 2016*). For logos expressing relative symbol frequency, background sequences are also aligned to the centroid. Logo computation then proceeds as usual, estimating the relative entropy between target and background sequences at each position in the alignment and the contribution of each symbol. Gaps introduced

in the centroid sequence are ignored, while gap symbols in the aligned sequences are treated as an additional symbol.

## Software availability

The *tcrdist3* code base used in this analysis is freely available at https://github.com/kmayerb/tcrdist3/ (copy archived at swh:1:rev:ecfc60a1569d656440c7fcfda841132451ad8b6e, *Mayer-Blackwell, 2021b*) with documented examples at https://tcrdist3.readthedocs.io/ relies on the Python package *pwseqdist* – freely available at https://github.com/agartland/pwseqdist (copy archived at swh:1:rev:d48d3bf4e6c79e5ba2417a1010f673179b27da68, *Fiore-Gartland, 2021*) – for numba-optimized just-in-time compiled versions of the TCRdist measure.

## Acknowledgements

This work was funded by NIH NIAID R01 AI136514-03 (PI Thomas) and ALSAC at St. Jude. The authors thank M Pogorelyy and A Minervina for extensive feedback on the manuscript. Scientific Computing Infrastructure at Fred Hutchinson Cancer Research Center was funded by ORIP grant S10OD028685.

## Additional information

### Competing interests

Jeremy C Crawford: JCC served as unpaid consultant for 10X Genomics on the initial analysis of the 10x_200k dataset. Tomer Hertz: TH has equity in Poold Diagnostics. Paul G Thomas: is on the Scientific Advisory Boards of Immunoscape and Cytoagents, consulted for Elevate Bio and PACT Pharma, and has received travel costs and speaking fees from 10X Genomics and Illumina. PT served as unpaid consultant for 10X Genomics on the initial analysis of the 10x_200k dataset. PT also has filed patents on methods for sequencing and cloning TCRs (International PCT applications published December 24, 2020 as WO 2020/257575 and January 7, 2021 as WO 2021/003114). These applications are pending and have not yet been granted. Philip Bradley: served as unpaid consultant for 10X Genomics on the initial analysis of the 10x_200k dataset. The other authors declare that no competing interests exist.

### Funding

| Funder | Grant reference number | Author |
| --- | --- | --- |
| National Institute of Allergy and Infectious Diseases | AI136514-03 | Koshlan Mayer-Blackwell<br>Stefan Schattgen<br>Jeremy C Crawford<br>Aisha Souquette<br>Jessica A Gaevert<br>Tomer Hertz<br>Paul G Thomas<br>Philip Bradley<br>Andrew Fiore-Gartland |
| National Institutes of Health | ORIP S10OD028685 | Koshlan Mayer-Blackwell<br>Andrew Fiore-Gartland |

The funders had no role in study design, data collection, and interpretation, or the decision to submit the work for publication.

### Author contributions

Koshlan Mayer-Blackwell, Conceptualization, Formal analysis, Methodology, Software, Visualization, Writing - original draft, Writing – review and editing; Stefan Schattgen, Liel Cohen-Lavi, Jeremy C Crawford, Aisha Souquette, Jessica A Gaevert, Conceptualization, Writing – review and editing; Tomer Hertz, Paul G Thomas, Philip Bradley, Conceptualization, Funding acquisition, Supervision, Writing – review and editing; Andrew Fiore-Gartland, Conceptualization, Formal analysis, Methodology, Software, Supervision, Visualization, Writing - original draft, Writing – review and editing

### Author ORCIDs

Koshlan Mayer-Blackwell http://orcid.org/0000-0002-1652-4023

Philip Bradley http://orcid.org/0000-0002-0224-6464
Andrew Fiore-Gartland http://orcid.org/0000-0001-7627-2166

**Decision letter and Author response**
Decision letter https://doi.org/10.7554/eLife.68605.sa1
Author response https://doi.org/10.7554/eLife.68605.sa2

## Additional files

**Supplementary files**
• Transparent reporting form
• Supplementary file 1. Supporting data and analysis results.

### Data availability

Source immune repertoire data is available through the immuneRACE project (https://immunerace. adaptivebiotech.com/data/).

The following previously published dataset was used:

| Author(s) | Year | Dataset title | Dataset URL | Database and Identifier |
|---|---|---|---|---|
| Nolan S, Vignali M, Klinger M, Dines JN, Kaplan IM, Svejnoha E, Craft T, Boland K, Pesesky M, Gittelman RM, Snyder TM, Gooley CJ, Semprini S, Cerchione C, Mazza M, Delmonte OM, Dobbs K, Carreño-Tarragona G, Barrio S, Sambri V, Martinelli G, Goldman JD, Heath JR, Notarangelo LD, Carlson JM, Martinez-Lopez J, Robins HS | 2020 | A large-scale database of T-cell receptor beta (TCRβ) sequences and binding associations from natural and synthetic exposure to SARS CoV-2 | https://immunerace. adaptivebiotech.com/ data/ | ImmuneCODE-MIRA-Release002, immunerace. adaptivebiotech.com/ data/ |

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
