## [Editor Report]

This paper introduces and validates a novel concept which will be of great interest to all those interested in T cell immunity and especially the T cell receptor repertoire. The concept builds on the idea that TCRs to the same antigen often share sequence similarities, which they quantify using a bespoke tool tcrdist3. Using this tool they develop the idea of a meta-clone, a set of TCRs sharing biochemical similarities and potentially recognising the same antigen. In this paper they further show that such clonotypes may show increased sharing between HLA-related individuals, and explore the use of such clonotypes in characterising antigen-specific immune response across cohorts of individuals.

---

## [Decision Letter]

**Decision letter after peer review:**

Thank you for submitting your article "TCR meta-clonotypes for biomarker discovery with tcrdist3 enabled identification of public, HLA-restricted, SARS-CoV-2 associated TCR features" for consideration by *eLife*. Your article has been reviewed by 3 peer reviewers, including Benny Chain as the Reviewing Editor and Reviewer #1, and the evaluation has been overseen by Aleksandra Walczak as the Senior Editor. The following individual involved in review of your submission has agreed to reveal their identity: Tahel Ronel (Reviewer #3).

The reviewers are all agreed that the content of the paper is of interest to the readership of the journal. However, they are also agreed that significant revisions are necessary to increase the impact of the paper, and we would like to invite you to submit a revised version of the manuscript.

Please address the detailed comments from all three reviewers and provide a point-by-point response. In particular, the emphasis of the paper needs to be shifted from COVID to the idea of the meta-clonotype. In order to achieve this, and to make the maximum impact, the paper must be thoroughly revised, to provide much more detail and clarity on the definition/construction of meta-clones, and how they can be used to define antigen-specific responses.

*Reviewer #1 (Recommendations for the authors):*

The authors should sharpen the manuscript to focus it exclusively on evaluating their new concept of a meta-clonotype, and not on evaluating the MIRA-based SARS-Cov-2 data set. For example, Lines 302-325 – what is the relevance of this section ? It analyses the MIRA dataset, and suggests the peptide-specific responses show HLA preference (hardly surprising) but doesn't say anything about the meta clonotypes.

The extension of TCRdist to gamma/delta cells seems irrelevant to this paper. No gamma-delta data are evaluated.

I found the section 560-574 (the details of generating the meta-clonotypes) very hard to follow. This is surely the crux of the whole paper, and the method for generating meta-clonotypes needs to be crystal clear. For example what does "With each candidate centroid, a meta-clonotype was engineered by selecting the maximum distance radius that still controlled the number of neighboring TCRs in the weighted unenriched background to 1 in 10^6^". How do you reach 10^6^ with only 200000 background TCRs?

Critically , in order to increase the impact of this study, it would be important to show that meta-clonotypes perform better than public clonotypes in identifying COVID-infected individuals (as done for clonotypes for CMV in Emerson et al. Nature Genetics 49,659).

*Reviewer #2 (Recommendations for the authors):*

Title: It's a bit strange to refer to meta-clonotypes as features (especially since machine learning was not performed). Also. the term feature does not have the same meaning throughout the manuscript. Can you adjust the title to be a bit more direct and less confusing?

Abstract: "As the mechanistic basis of adaptive cellular antigen recognition, T cell receptors (TCRs) encode" → this sentence doesn't make sense. There are many mechanisms involved in immune recognition (binding, proliferation etc…), TCR sequences are certainly a part of it but I would hardly call them the "mechanistic basis" – can you rephrase? I would like the term "mechanistic" to go – this paper is not mechanistic in any way.

"17 SARS-CoV-2 antigen-enriched repertoires" → antigen-enriched doesn't mean what you think it means. Can you please rephrase throughout the manuscript? What you mean is antigen-annotated/antigen-specific/etc…

Introduction: The introduction seems a bit verbose (the entire paper is quite verbose…more streamlining by making text and captions more precise would greatly enhance readability). This is not a covid-centric paper – please dramatically reduce the SARS-CoV-2 section. Also, the first paragraph can be written in 5 instead of 35 lines. Please try to get closer to one page overall.

Results Figure 1A: what does "searchable public meta-clonotype" mean? It's not mentioned anywhere else in the text.

I think it would be much more useful to the reader if you illustrate how tcrdist calculates tdus, how they are to be interpreted (you mention in the text: 1 aa mismatch is 12 tdus – this is great and would be nice to see in Figure 1A for example).

Figure 1B In my opinion, the figure does not reflect what you want it to say (quantification of the frequency of putative meta-clonotypes). Please adjust the figure accordingly.

Figure 2A "As the radius about a TCR centroid expands, the number of TCRs it encompasses naturally increases; the rate of increase is more rapid in the antigen-enriched 167 repertoires compared to the unenriched repertoires" → can you quantify this rate? How does the rate depend on sequencing depth? Can you also add the OLGA-generated data to this plot? Is your assumption that the OLGA-generated data would perform close to that of the cord blood data?

For all figures: can you change the bold text of the caption to the main result of the figure. As of now, the bold text doesn't really say much.

If I do a text search in the main text for "Dash BMLF", I don't find anything. Please define all named datasets in the methods/main text.

Regarding the antigen-specific data – is the higher rate in antigen-specific data due to the fact that you are focusing on only a few peptides here? Would the rate be similar to the baseline data if you somehow normalized for that? Like for example only taking one tcr per peptide? Probably not possible, right, given the sparsity of the data..?

How does Tcrdist3 normalize for length differences? Do these curves differ across tcr length, germline genes?

Figure 2B

Can you explain in the main text and the methods what MIRA M48 means –specifically, what do each of the numbers mean?

Figure 3

"This suggests that TCRs within sparse neighborhoods represent less common modes of antigen recognition and highlights the broad heterogeneity of neighborhood densities even among TCRs recognizing a single pMHC." → to what extent can TCRs with sparse neighborhood be a result of undersampling?

Can you add OLGA-generated data to Figure 3 as well?

Figure 4

Can you mention in the caption how many TCRs you are investigating in this subfigure? Please also check for all other figures where relevant.

"We also noted that TCRs with 191 empty neighborhoods tended to have longer CDR3 loops (Figure 4C)" Where and how do I see this in Figure 4C?

"To be useful, a meta-clonotype definition should be broad enough 206 to capture multiple biochemically similar TCRs" → what do you mean by biochemically similar? I think, in the AIRR field, when people speak of biochemically similar, they mean something related to Atchley/Kidera factors. I don't think this is what you mean here, right? Maybe rephrase to avoid misunderstandings?

"This is similar to previous approaches taken by tools like ALICE and TCRNET, except that we employ a biochemically informed distance measure (TCRdist)" → this statement is a bit indirect for my taste. Can you rephrase and make it really clear in what respect tcrdist3 differs from Alice et al.?

How did you decide on the number of 100000 IGOR and cord blood TCRs?

"One part consisted of 100,000 synthetic TCRs whose TRBV- and TRBJ-gene frequencies matched those in the antigen-enriched repertoire; TCRs were generated using the software OLGA" → how did you make sure that frequencies matched?

"Using this approach, we are able to estimate the abundance of TCRs similar to a centroid TCR in an unenriched background repertoire of effectively ~1,000,000 TCRs," → Where does the number of 1M TCRs come from? Can you show that calculations don't change if you sample another 100000? To what extent do you think it plays a role that the cord blood data has been generated with a completely different experimental protocol than the MIRA data?

Figure 5

Can you please add to "HLA genotype inferences" section in Methods the prediction accuracies for HLA-B alleles used in Figure 5A? If prediction accuracies are low, please remove those data also from Figure 5A.

Is the HLA-classifier publicly available? Does a package for it exist as well? Which HLA alleles other than those mentioned are quite safe to predict from sequencing data?

Independently of Figure 5, is there a Figure where I see how much meta-clonotypes make up of a repertoire in terms of sequences and sequencing reads? Basically, how much more of the antigen-specific portion of a repertoire does one capture if looking at meta-clonotypes?

Furthermore, how do you compare meta-clonotypes across individuals (publicity)? Can a TCR be part of several meta-clonotypes? Can you explain all of this more in Figure 1 and the main text?

Figure 7

To what extent is this figure needed in the main text?

Since your approach is actually quite similar to Alice, especially, since you include in your analysis pgens, wouldn't it be more interesting to relate in depth to Alice than to GLIPH?

GLIPH2 was used with the default TCR background (line 629). Would this give the TCRdist3 meta-clonotypes an advantage, since the used background for TCRdist3 has been more densely sampled around the biochemical neighborhoods of interest (line 563)?

In the comparison to k-mer based CDR3 features, were meta-clonotypes defined by RADIUS or RADIUS +MOTIF? Please also mention this in Figure 7.

More general comments:

The paper presents meta-clonotypes as a novel approach to comparing similar TCRs across repertoires and mainly compares the meta-clonotypes approach to the use of public exact TCRs. This comparison seems a bit trivial since any sequence similarity clustering approach is expected to perform better than exact TCR matches. I think it's very interesting to show that meta-clonotype spaces differ between antigen-specific and non-specific repertoires. It would have been nice to focus the analysis more on this and to actually discover something cool about the repertoire biology than trying to relate exclusively to covid.

How does your method compare to this recently published approach based on TCR sub-repertoires shared across individuals? https://bmcbioinformatics.biomedcentral.com/articles/10.1186/s12859-021-04087-7

*Reviewer #3 (Recommendations for the authors):*

The methodology proposed in this manuscript (tcrdist3) has been made publicly available through GitHub. The application to COVID-19 is based on public datasets and is clearly referenced. Data derived in the analysis, such as NetMHCpan predictions and the set of derived meta-clonotypes are included as supplementary material, which is helpful and adheres to *eLife*'s policies.

The dataset of derived COVID-19 related meta-clonotypes is a valuable resource for the analysis of other bulk repertoire COVID-19 datasets, and the proposed method should be applicable in a variety of antigenic settings. This dataset could be further characterised, perhaps as a supplementary/additional figure: the distribution of optimal radii, distribution of number of TCRs conforming to each meta-clonotype, number of people contributing TCRs to the meta-clonotype, are these different between the strong HLA meta-clonotypes and weak HLA meta-clonotypes? This would help when applying the method in other contexts.

[Editors' note: further revisions were suggested prior to acceptance, as described below.]

Thank you for resubmitting your work entitled "TCR meta-clonotypes for biomarker discovery with *tcrdist3* enabled identification of public, HLA-restricted clusters of SARS-CoV-2 TCRs" for further consideration by *eLife*. Your revised article has been reviewed by 3 peer reviewers, one of whom is a member of our Board of Reviewing Editors, and the evaluation has been overseen by Aleksandra Walczak as the Senior Editor.

The reviewers agree that the manuscript has been considerably improved but there are a few remaining issues of clarity that need to be addressed, as outlined below by reviewer 3.

Specifically:

1. Please include a clear and consistent definition of a meta-clone in the Discussion. If in fact there are multiple alternative definitions, please clearly set these out, with an indication of when each would be used.

2. Clearly indicate which background set is used in each figure/section of the paper.

3. Further sharpen the Discussion around comparing the meta-clonotype approach to other existing methods. Specifically, please clarify the relative importance to the novel meta-clonotype defining which derives from the new TCRdist3 metric, the motif, and the novel approach to establishing background comparisons.

The comments of the reviewers are listed below.

*Reviewer #2 (Recommendations for the authors):*

The authors have addressed all of my comments.

*Reviewer #3 (Recommendations for the authors):*

The authors have placed more focus in the revised manuscript on the definition and generation of meta-clonotypes, as suggested, with the covid data used as an example application. They have included a couple of new analyses to this effect (background sets 'sensitivity analysis', logo sequence characterisation). While I still think that the idea of meta-clonotypes is both interesting and potentially useful, I find some of the paper a bit long and difficult to follow.

For example, it looks to me like the definition of meta-clonotype changes along the paper: e.g. In Figure 1/Figure 6 a meta-clonotype is centroid (TRBV + CDR3) + radius +/- motif; in Figure 10 it also includes what I think is an identifier for the set it comes from; and in Figure 12 it is TRBV + TRBJ + CDR3 + radius. As this is the main focus of the paper I think this definition should be made clear, consistent and explained somewhere.

Secondly, it was unclear to me when the authors use which set for background: e.g. does Figure 1 and caption 'synthetic set' refer to the set of 100k OLGA V-J biased + 100k cord used later? cf Lines 535-537 "background CDR3s that were sampled from cord blood and constrained to use the same V and J genes", and elsewhere unadjusted cord blood is used without the OLGA adjusted set.

I also think that the advantage / unique usage of the proposed meta-clonotype method over existing methods for grouping antigen-specific TCRs should be further clarified and emphasised: I personally don't find the Results section 'Comparison to k-mer based CDR3 features' or Figures 11 and 12 very convincing to this effect, and think this message should be made clearer in the paper before the sentence in the discussion "Our framework is designed for a different task than these algorithms…".

I do think the meta-clonotype method is useful, these changes are addressable and the paper has the potential to be made more readable and more impactful, if the authors streamline the definition and explanations, remove repetitive sections, and cross-reference to the relevant places in the text where things are referred to in the text before their explanations.

---

## [Author Response]

Reviewer #1 (Recommendations for the authors):The authors should sharpen the manuscript to focus it exclusively on evaluating their new concept of a meta-clonotype, and not on evaluating the MIRA-based SARS-Cov-2 data set. For example, Lines 302-325 – what is the relevance of this section ? It analyses the MIRA dataset, and suggests the peptide-specific responses show HLA preference (hardly surprising) but doesn't say anything about the meta clonotypes.

We agree with you and understand why you think it’s important to focus on the concept of the meta-clonotype and have revised the manuscript towards this goal. Lines 333-337 help to establish prior evidence of HLA restriction for a subset of the MIRA datasets, which becomes critical later when we provide evidence of consistent HLA restriction of meta-clonotype conforming TCRs in COVID-19 patients; it’s this consistency that helps demonstrate the value of meta-clonotypes over individual clonotypes. However, we recognized that HLA analysis section in our original version distracted from our main point, so we have shortened this section in the main text.

The extension of TCRdist to gamma/delta cells seems irrelevant to this paper. No gamma-delta data are evaluated.

We agree that this is not a focus of the paper, however we included this information as the ability to analyze gamma/delta TCRs differentiate tcrdist3 from other currently available tools. The ability to perform analysis on paired-chain data as well as gamma/delta TCRs are important features of the software and therefore we’ve left one brief reference to these features in the Discussion section.

I found the section 560-574 (the details of generating the meta-clonotypes) very hard to follow.

Thank you for bringing this to our attention. We have substantially revised the Results paragraph that describes how meta-clonotypes are generated, including the radius selection procedure (lines 178-258). The additions include a formula (line 235) for implementing the adjustment required for the sampling design and a sensitivity analysis (lines 259-294) supporting the choice of background size and make-up. With all details in the Results, we have removed this section from the Methods.

This is surely the crux of the whole paper, and the method for generating meta-clonotypes needs to be crystal clear. For example what does "With each candidate centroid, a meta-clonotype was engineered by selecting the maximum distance radius that still controlled the number of neighboring TCRs in the weighted unenriched background to 1 in 10^6^". How do you reach 10^6^ with only 200000 background TCRs?

This is achieved by sampling background TCRs from regions of “TCR space” that are proximal to the centroid, based on VJ-gene matching. This greatly increases the efficiency of the background comparison since the vast majority of TCR space is not relevant and can be more sparsely sampled. We then use weighting to account for this sampling bias; previously we referred to this technique as inverse probability weighting (IPW), and while the concept is derived from IPW we think it’s more clearly communicated without this statistical terminology, and therefore have removed this from the manuscript. This is explained in greater detail in the Results section, “Meta-clonotype radius can be tuned to balance sensitivity and specificity” on lines 178-258.

Critically , in order to increase the impact of this study, it would be important to show that meta-clonotypes perform better than public clonotypes in identifying COVID-infected individuals (as done for clonotypes for CMV in Emerson et al. Nature Genetics 49,659).

While it may seem that building a repertoire-based classifier for SARS-CoV-2 infection would be a great way to demonstrate the value of meta-clonotypes, there are good reasons not to include such an analysis in the manuscript. One reason is that we would not necessarily expect that meta-clonotypes would be better than public clonotypes for identifying SARS-CoV-2 infected individuals. In fact, others have shown (Snyder et al., 2020) that with a massive dataset of cases and controls it is possible to identify a relatively small number of individual clones that are capable of identifying individuals with prior infection. Instead, we propose that meta-clonotypes would be more useful than individual clonotypes when, (a) less case-control data is available to train a classifier, (b) focus on a specific epitope is required, or (c) when it would be useful to gain insight into the underlying biology (e.g., epitope immunodominance hierarchies, response polyclonality, or CDR3 motifs defining antigen specificity). We also note that the uninfected control dataset that enabled the above analysis at the clonotype level has not been made publicly available, preventing us and others from making further improvements. Lastly, we agree with previous comments that this manuscript would benefit from a sharpened focus on the concept of the meta-clonotype. Therefore, we have chosen not to make efforts to build a classifier for prior infection and instead have emphasized the existing application, which demonstrates how meta-clonotypes retain antigen-specificity and how they can be used for population-level analysis.

Reviewer #2 (Recommendations for the authors):Title: It's a bit strange to refer to meta-clonotypes as features (especially since machine learning was not performed). Also. the term feature does not have the same meaning throughout the manuscript. Can you adjust the title to be a bit more direct and less confusing?

We see how referring to meta-clonotypes as features may be confusing and have revised the title accordingly: “TCR meta-clonotypes for biomarker discovery with tcrdist3 enabled identification of public, HLA-restricted clusters of SARS-CoV-2 TCRs”

Abstract: "As the mechanistic basis of adaptive cellular antigen recognition, T cell receptors (TCRs) encode" → this sentence doesn't make sense. There are many mechanisms involved in immune recognition (binding, proliferation etc…), TCR sequences are certainly a part of it but I would hardly call them the "mechanistic basis" – can you rephrase? I would like the term "mechanistic" to go – this paper is not mechanistic in any way.

We agree and removed that phrasing.

"17 SARS-CoV-2 antigen-enriched repertoires" → antigen-enriched doesn't mean what you think it means. Can you please rephrase throughout the manuscript? What you mean is antigen-annotated/antigen-specific/etc…

We appreciate that the term “antigen-enriched” may not mean the same thing to everyone. We have adopted the term antigen-associated TCRs to refer to a set of TCRs that have an overrepresentation of those that recognize a particular antigen. We have provided an explicit definition of this usage in the Introduction and use the term throughout.

Introduction: The introduction seems a bit verbose (the entire paper is quite verbose…more streamlining by making text and captions more precise would greatly enhance readability). This is not a covid-centric paper – please dramatically reduce the SARS-CoV-2 section. Also, the first paragraph can be written in 5 instead of 35 lines. Please try to get closer to one page overall.

We appreciate the need for brevity. The Introduction has been streamlined and any background on COVID-19 that was not directly related to the analyses was removed.

Results Figure 1A: what does "searchable public meta-clonotype" mean? It's not mentioned anywhere else in the text.

This phrase has been replaced with “Public meta-clonotype”: we had meant that the meta-clonotype was something that could be used to search for conformant sequences in bulk repertoires, but this point is now clarified elsewhere.

I think it would be much more useful to the reader if you illustrate how tcrdist calculates tdus, how they are to be interpreted (you mention in the text: 1 aa mismatch is 12 tdus – this is great and would be nice to see in Figure 1A for example).

We agree that having an intuition for tdus is important to understanding the definition of a meta-clonotype and its radius. We have added a new Figure 2 that plots the distances between illustrative pairs of antigen-associated TCR CDR3s (MIRA55) using units of edit-distance versus TCR distance units.

Figure 1B In my opinion, the figure does not reflect what you want it to say (quantification of the frequency of putative meta-clonotypes). Please adjust the figure accordingly.

The Figure 1B caption now begins: “Quantifying meta-clonotype conformant TCRs in a bulk repertoire” and Figure 1C “Population-level analysis of TCR meta-clonotype frequency”

Figure 2A "As the radius about a TCR centroid expands, the number of TCRs it encompasses naturally increases; the rate of increase is more rapid in the antigen-enriched 167 repertoires compared to the unenriched repertoires" → can you quantify this rate? How does the rate depend on sequencing depth? Can you also add the OLGA-generated data to this plot? Is your assumption that the OLGA-generated data would perform close to that of the cord blood data?

Figure 2 is now Figure 3. What we meant here was to make the observation that the proportion of TCRs with a neighbor within a set of antigen-associated TCRs was greater than for TCRs among a set that has not been experimentally enriched for T cells recognizing a single antigen (e.g., cord blood), across a range of neighborhood radii. Indeed, the size of the set of TCRs does affect the proportion of TCRs with at least one neighbor, which is why we chose to include cord blood repertoires with 1K and 10K TCRs each; the bigger a set of TCRs the more likely it is to find at least one that is a neighbor. As we push this observation further in the manuscript, it’s helpful to consider the proportion of TCRs included in each TCR’s neighborhood rather than the proportion with at least one neighbor; this is what we plot in Figure 3 (now revised Figure 4), again for antigen-associated and cord blood TCRs. The goal of Figure 2 (now revised Figure 3) is to start with a more intuitive observation and then move to Figure 3 (now revised Figure 4) which is a better way to quantify the observation of neighborhood density as a function of radius; the statistic plotted in Figure 3 (now revised Figure 4) for each TCR, “proportion of TCRs within a specified radius” is also now robust to the set size compared to a simple indicator of whether or not a neighbor is present.

We have conducted several sensitivity analyses evaluating cord blood, OLGA sequences and a mix of the two, as background repertoires for selecting the optimal radius for each meta-clonotype (lines 259-294). These analyses show efficiently sampled OLGA sequences behave similarly to cord blood sequences when estimating meta-clonotype neighborhood densities. We concluded that additional lines on Figure 3A for OLGA repertoires of comparable sizes wouldn’t help make the intended point that neighborhoods around antigen-associated TCRs are more densely populated than those around individual TCRs in an unselected repertoire (e.g. cord blood). We have added a sub-panel with ECDFs from OLGA generated TCRs in the new Figure 4D.

For all figures: can you change the bold text of the caption to the main result of the figure. As of now, the bold text doesn't really say much.

The first line of the figure captions have been revised to be more descriptive.

If I do a text search in the main text for "Dash BMLF", I don't find anything. Please define all named datasets in the methods/main text.

The Dash et al. publicly available source of these tetramer-sorted TCRs is now included in the figure caption.

Regarding the antigen-specific data – is the higher rate in antigen-specific data due to the fact that you are focusing on only a few peptides here? Would the rate be similar to the baseline data if you somehow normalized for that? Like for example only taking one tcr per peptide? Probably not possible, right, given the sparsity of the data..?

We’re not sure we fully understand your point, but will try to offer further clarification. We think you are making the point that the size of the TCR set matters. For instance, you could take one TCR from either an antigen-associated set or a background set and count how many neighbors it has within its set for a given radius. The number tallied would depend on the size of the set AND on the density of the neighborhood. We show 1K and 10K cord blood repertoires because it shows that as the set size decreases there are fewer TCRs in the neighborhood. If we used 100 or 500 TCRs, which would be similar to some of the antigen-associated sets, it would probably result in an even flatter line; we included 1K and 10K because they are conservatively larger than the antigen-associated sets and yet are still generally flatter lines. Also, keep in mind that the line represents an average across all TCRs in the set, which will also be more variable for smaller sets. We’ve now noted this last point of clarification in the Figure 3 caption.

How does Tcrdist3 normalize for length differences? Do these curves differ across tcr length, germline genes?

The TCRdist metric penalizes differences in CDR3 length as a non-conservative amino-acid substitution; this has been clarified in the methods. We show in Figures 3 and 4 (now Figures 4 and 5) that neighborhood density is related to CDR3 length by plotting curves by centroid generation probability. TCRs with lower probability of generation typically have fewer neighbors in the background. We have not systematically studied different VJ-genes but their usage in a background repertoire would have a similar effect on neighborhood density.

Figure 2BCan you explain in the main text and the methods what MIRA M48 means – specifically, what do each of the numbers mean?

This is an important point so we have pulled the definition from the Methods and added a definition in the Introduction: “we refer to these sets as MIRA1 through MIRA252 in rank order by their size” (Lines 95-96).

Figure 3"This suggests that TCRs within sparse neighborhoods represent less common modes of antigen recognition and highlights the broad heterogeneity of neighborhood densities even among TCRs recognizing a single pMHC." → to what extent can TCRs with sparse neighborhood be a result of undersampling?

If we make an assumption that sampling is uniform across the repertoire space, then, while the absolute number of neighbors may be affected by under/over sampling, the relative number of neighbors for one TCR compared to another should be controlled for within a set of antigen-associated TCRs. It’s possible that near the level of detection (e.g. frequencies near 1/n) the presence of a neighbor may be observed in this experiment, but not if we had repeated the experiment (i.e. high variability), but it shouldn’t be biased to underestimate or overestimate density. Therefore, we believe that the heterogeneity we see within each MIRA set represents real variation in neighborhood density.

Can you add OLGA-generated data to Figure 3 as well?

Yes. We have added OLGA-generated data to revised Figure 3D, now Figure 4D.

Figure 4Can you mention in the caption how many TCRs you are investigating in this subfigure? Please also check for all other figures where relevant.

Yes, added and addressed throughout.

"We also noted that TCRs with 191 empty neighborhoods tended to have longer CDR3 loops (Figure 4C)" Where and how do I see this in Figure 4C?

We have clarified this observation in the text. Revised Figure 4C shows that TCRs with sparse or empty neighborhoods tend to have longer CDR3 loops.

"To be useful, a meta-clonotype definition should be broad enough 206 to capture multiple biochemically similar TCRs" → what do you mean by biochemically similar? I think, in the AIRR field, when people speak of biochemically similar, they mean something related to Atchley/Kidera factors. I don't think this is what you mean here, right? Maybe rephrase to avoid misunderstandings?

While a sequence-based distance could simply count substitutions, with TCRdist we are using a substitution matrix (BLOSUM62) that down-weights substitutions of biochemically similar amino acids. It also upweights the CDR3 because of its importance in making contacts with the antigen. Therefore, we think that referring to TCRdist defined neighbors as biochemically similar is appropriate.

"This is similar to previous approaches taken by tools like ALICE and TCRNET, except that we employ a biochemically informed distance measure (TCRdist)" → this statement is a bit indirect for my taste. Can you rephrase and make it really clear in what respect tcrdist3 differs from Alice et al.?

Yes, please see response above. A complete comparison of these methods now appears in the Discussion.

How did you decide on the number of 100000 IGOR and cord blood TCRs?"One part consisted of 100,000 synthetic TCRs whose TRBV- and TRBJ-gene frequencies matched those in the antigen-enriched repertoire; TCRs were generated using the software OLGA" → how did you make sure that frequencies matched?"Using this approach, we are able to estimate the abundance of TCRs similar to a centroid TCR in an unenriched background repertoire of effectively ~1,000,000 TCRs," → Where does the number of 1M TCRs come from? Can you show that calculations don't change if you sample another 100000? To what extent do you think it plays a role that the cord blood data has been generated with a completely different experimental protocol than the MIRA data?

Please see response above. We have conducted sensitivity analyses studying the impact of the size of the background on meta-clonotype radius optimization, which appear in the revised manuscript. To your specific questions that are not answered above: V and J gene frequencies of the OLGA set were matched to their frequencies in the cord blood dataset. Also, OLGA was modified to generate synthetic TCRs with specific V and J genes (this modified version is available on GitHub). We don’t know to what extent the technical factors play a role in sampling from the background and using it as a comparator for the MIRA data. Ideally, one could generate a background that is relevant to the experiment and employ identical sequencing protocols. We think its promising that the OLGA (after VJ-gene matching) and cord blood background performed consistently. Ultimately, we think the choice of background should be informed by the experiment, including possible technical factors. We’ve added this commentary to the manuscript.

Figure 5Can you please add to "HLA genotype inferences" section in Methods the prediction accuracies for HLA-B alleles used in Figure 5A? If prediction accuracies are low, please remove those data also from Figure 5A.Is the HLA-classifier publicly available? Does a package for it exist as well? Which HLA alleles other than those mentioned are quite safe to predict from sequencing data?

Yes, this classifier has been made available on GitHub (github.com/kmayerb/hla3). Prediction accuracies and sensitivity vary by allele, but generally sensitivity ranged between 0.85-0.97 and specificity 0.97-1 for the major HLA-A and HLA-B alleles we have analyzed here: HLA-A*01:01, HLA-A*02:01, HLA-A*03:01, HLA-A*24:02, HLA-A*11:01, HLA-B*07:02, HLA-B*15:01, HLA-B*35:03, HLA-B*57:01, HLA-B*40:01.

Independently of Figure 5, is there a Figure where I see how much meta-clonotypes make up of a repertoire in terms of sequences and sequencing reads? Basically, how much more of the antigen-specific portion of a repertoire does one capture if looking at meta-clonotypes?

This is an important question, but we do not know the answer. With something like a tetramer-sorted dataset we can answer that question and show what proportion of the antigen-specific sequences are captured by meta-clonotypes. We are essentially doing this with the MIRA ECDFs in Figures 3, 4, and 5 (now Figures 4,5,6), but this is circular because the meta-clonotypes were of course defined to identify these TCRs. Perhaps with a large hold-out dataset one could look to see how well these meta-clonotypes perform; experimentally one might want to generate large tetramer-sorted datasets from many individuals to show that the meta-clonotypes work in a population. With the COVID-19 analysis we have attempted to demonstrate this statistically, and while it doesn’t provide an estimate of how much of the antigen-specific repertoire we’re capturing, it does show that the antigen-specific signal is statistically significant.

Furthermore, how do you compare meta-clonotypes across individuals (publicity)? Can a TCR be part of several meta-clonotypes? Can you explain all of this more in Figure 1 and the main text?

Yes, see response above about overlapping meta-clonotypes.

Figure 7To what extent is this figure needed in the main text?

Figure 7 is now Figure 11. This figure compares meta-clonotype to k-mere based features for 16 MIRA sets.

Since your approach is actually quite similar to Alice, especially, since you include in your analysis pgens, wouldn't it be more interesting to relate in depth to Alice than to GLIPH?

Our impression is that GLIPH2 is a more commonly used tool (due to computational performance) for finding potentially interesting groups of TCRs and the feedback we’ve received is that it would be helpful to compare the two in the manuscript. Therefore, we have kept Figure 7 (now Figure 11) in the manuscript. To clarify, we do not explicitly use Pgen in forming meta-clonotypes; instead we are estimating the likelihood of observing a meta-clonotype conformant TCR in the background empirically. As we understand it, ALICE would not be applicable to this context when trying to find public features among TCRs that have already been enriched for sharing an antigen specificity; this enrichment would make most if not all groupings seem significantly unexpected in an ALICE model.

GLIPH2 was used with the default TCR background (line 629). Would this give the TCRdist3 meta-clonotypes an advantage, since the used background for TCRdist3 has been more densely sampled around the biochemical neighborhoods of interest (line 563)?

The GLIPH2 default background is large (CD8, 573,211 clones, CD4 772,312 clones), which may actually give GLIPH2 the advantage. Instead, we use a smaller background that has greater efficiency, due to the dense sampling of biochemically similar TCRs. We decided that using the default GLIPH2 background is the most relevant for comparison since that’s what most people seem to use.

In the comparison to k-mer based CDR3 features, were meta-clonotypes defined by RADIUS or RADIUS+MOTIF? Please also mention this in Figure 7.

The comparisons were based on RADIUS+MOTIF meta-clonotypes. This has been clarified in Figure 7 (now Figure 11).

More general comments:The paper presents meta-clonotypes as a novel approach to comparing similar TCRs across repertoires and mainly compares the meta-clonotypes approach to the use of public exact TCRs. This comparison seems a bit trivial since any sequence similarity clustering approach is expected to perform better than exact TCR matches. I think it's very interesting to show that meta-clonotype spaces differ between antigen-specific and non-specific repertoires. It would have been nice to focus the analysis more on this and to actually discover something cool about the repertoire biology than trying to relate exclusively to covid.

We’ve revised the manuscript to show how meta-clonotypes can be to form interesting biological hypotheses about antigen specificity from experiments that enrich antigen-specific TCRs. We also think it’s important that meta-clonotypes can be used for population-level statistics and biomarker development, a necessary step towards clinical translation. Still we’ve de-emphasize the COVID-19 analysis, focusing on how it demonstrates the performance of meta-clonotypes.

How does your method compare to this recently published approach based on TCR sub-repertoires shared across individuals? https://bmcbioinformatics.biomedcentral.com/articles/10.1186/s12859-021-04087-7

Yohannes et al. 2021 recently published an interesting methodology. Briefly, Yohannes and colleagues use 4-nt-mers or 3-aa-mers to cluster sets of CDR3 sequences within a sample from a single TCR repertoire. In the 3-mer aa amino acid case, a (20^3^) 8000-dimension vector represents each CDR3s. Within-repertoire diversity is reduced by clustering vectors via agglomerative clustering. Mean valued vector represents each cluster (which the authors call a centroid). Centroid verctors from multiple samples are clustered across samples to identify clusters representing common repertoire features. This novel approach was applied to a small cohort of patients with Celiac disease and was able to identify clusters of similar but non-identical TCRs across samples that were similar to antigen-associated TCRs from prior studies. This and other k-mer methods (see also Grief et al. 2017) and GLIPH (Glanville 2017) and GLIPH2 (Huang 2020), have shown that k-mer representation of CDR3s encode information that generalizes across samples.

The crucial distinction is that these bottom up methods learn enriched k-mer features within a single repertoire and then seek to see if those patterns are found in multiple samples. By contrast, the methodology we describe attempts to leverage information gained from experimental identification of antigen-associated TCRs from many samples to identify more antigen-associated TCRs in bulk data. We have sought to make this distinction clear in the revised manuscript, in particular line 53-65, 483-497.

Reviewer #3 (Recommendations for the authors):The methodology proposed in this manuscript (tcrdist3) has been made publicly available through GitHub. The application to COVID-19 is based on public datasets and is clearly referenced. Data derived in the analysis, such as NetMHCpan predictions and the set of derived meta-clonotypes are included as supplementary material, which is helpful and adheres to eLife's policies.The dataset of derived COVID-19 related meta-clonotypes is a valuable resource for the analysis of other bulk repertoire COVID-19 datasets, and the proposed method should be applicable in a variety of antigenic settings. This dataset could be further characterised, perhaps as a supplementary/additional figure: the distribution of optimal radii, distribution of number of TCRs conforming to each meta-clonotype, number of people contributing TCRs to the meta-clonotype, are these different between the strong HLA meta-clonotypes and weak HLA meta-clonotypes? This would help when applying the method in other contexts.

These are interesting points. We provide characteristics of the other MIRA sets in supporting Table S6 including the number of unique TCRs contributing to each meta-clonotype and the number of participants contributing to each meta-clonotype. The additional sensitivity analyses focused on radius optimization now show the distribution of radii across meta-clonotypes; radius and the number of subjects contributing to each meta-clonotype is also included in Tables S7 and S8. In general, the highest quality meta-clonotypes include many input TCRs and are highly public.

[Editors' note: further revisions were suggested prior to acceptance, as described below.]

The reviewers agree that the manuscript has been considerably improved but there are a few remaining issues of clarity that need to be addressed, as outlined below by reviewer 3.Specifically:1. Please include a clear and consistent definition of a meta-clone in the Discussion. If in fact there are multiple alternative definitions, please clearly set these out, with an indication of when each would be used.

We appreciate the opportunity to further refine the manuscript. We agree that clarity on the definition of a meta-clonotype is critical. As a result, we have moved a clear description to the introduction; technical discussion of the definition and its contrast with other TCR groupings is in the Discussion, with greater clarity.

We define a meta-clonotype to be a grouping of biochemically similar clonotypes surrounding a centroid clonotype, which enables population-level analyses that can consider all members of the meta-clonotype as a functional unit. In this study we developed a framework for defining meta-clonotypes from antigen associated TCRs based on a similarity threshold (i.e., a radius) and optionally, a CDR3 motif, which represent a portable set of criteria that can be used to identify meta-clonotype conformant TCRs across datasets. As experimental and statistical approaches evolve, we welcome new methods for defining TCR meta-clonotypes, and hope that the new term and concept continues to be useful, in so much that it captures the need for defining groups of functionally similar clonotypes for analysis. See lines 53-80 for clarification of this definition in the manuscript.

2. Clearly indicate which background set is used in each figure/section of the paper.

Throughout the manuscript, the background used to generate meta-clonotypes refers to the following, which we now state explicitly:

“A synthetic background was generated using 100,000 OLGA-generated TCRs and 100,000 TCRs sub-sampled from umbilical cord blood; OLGA-generated TCRs were sampled to match the V-J gene frequency in each MIRA receptor set, with weighting to account for the sampling bias (see Methods for details).”

In one separate instance, a different type of “background” is used to create motif visualization as shown as Figure 10. In Figure 10, the “background” used to refers to a background-subtracted CDR3 logo motif. Here the background is a random sampling of TCR beta receptors from cord blood with the same TRBV and TRBJ gene usage frequency as the receptors found within a TCRdist 20 radius of meta-clonotype centroid TRBV28*01 CASSLKTDAYEQYF.

To ensure clarity, we have modified figure captions with explicit descriptions of the backgrounds:

– We have added this description to Figure 1A caption to match wording in Figure 6:

“A synthetic background was generated using 100,000 OLGA-generated TCRs and 100,000 TCRs sub-sampled from umbilical cord blood; OLGA-generated TCRs were sampled to match the V-J gene frequency in each MIRA receptor set, with weighting to account for the sampling bias (see Methods for details).”

– Figure 4C and 4D, respectively shows proportion of neighbors in 9866 OLGA and 10,000 Cord Blood receptors as now specified in the caption text.

– Figure 7 involves sensitivity testing varying background size and type as specified in the caption.

– Figure 8: we added the caption text for clarity:

“Meta-clonotype radii were engineered using synthesized backgrounds developed for each MIRA set. Each background contained 100,000 OLGA-generated TCRs and 100,000 TCRs sub-sampled from umbilical cord blood; OLGA-generated TCRs were sampled to match to the V-J gene frequency in each MIRA receptor set (i.e., MIRA 1, 10, 30, 44,45, 48, 51, 53, 55, 70, 99, 110, 111,118, 133, 140, or 183) with weighting to account for the sampling bias (see Methods for details).”

– Figure 10: The background-adjusted logos are constructed by randomly sampling TCR β receptors from cord blood with the same TRBV and TRBJ gene usage, with 100 V-J matched TCRs sampled for every receptor in the foreground set.

– Figure 13: we added the caption text for clarity:

“Meta-clonotype radii were engineered using synthesized backgrounds developed for each MIRA set. Each background contained 100,000 OLGA-generated TCRs and 100,000 TCRs sub-sampled from umbilical cord blood; OLGA-generated TCRs were sampled to match to the V-J gene frequency in each MIRA receptor set with weighting to account for the sampling bias (see Methods for details).”

3. Further sharpen the Discussion around comparing the meta-clonotype approach to other existing methods. Specifically, please clarify the relative importance to the novel meta-clonotype defining which derives from the new TCRdist3 metric, the motif, and the novel approach to establishing background comparisons.

We agree that its critical to provide clarity about how this framework to develop antigen-associated meta-clonotypes is different from other published methods of grouping TCRs and also about when meta-clonotypes are a suitable tool for analysis. The fundamental difference is that other tools like GLIPH2, TCRNET and ALICE were not developed to find groups of similar TCRs among TCRs that have been experimentally pre-enriched for antigen-specificity. We’ve included a paragraph in the Discussion (quoted below), which emphasizes this distinction and the consequence for analyses.

We also address how the TCRdist radius, the motif and the novel approach to efficient background comparisons all contribute to differentiating our meta-clonotype framework from existing methods. In the revised manuscript, we write:

“The meta-clonotype framework we present joins a class of commonly used methods for TCR analysis that depend on comparisons to an antigen-naïve background repertoire. […] Ultimately, it is the focus on controlling the absolute frequency of meta-clonotype conformant TCRs in an antigen-naïve background that gives the meta-clonotype definitions portability to be applied to analyses of bulk repertoires, where quantification of similar antigen-specific TCRs is required.”

Reviewer #3 (Recommendations for the authors):The authors have placed more focus in the revised manuscript on the definition and generation of meta-clonotypes, as suggested, with the covid data used as an example application. They have included a couple of new analyses to this effect (background sets 'sensitivity analysis', logo sequence characterisation). While I still think that the idea of meta-clonotypes is both interesting and potentially useful, I find some of the paper a bit long and difficult to follow.For example, it looks to me like the definition of meta-clonotype changes along the paper: e.g. In Figure 1/Figure 6 a meta-clonotype is centroid (TRBV + CDR3) + radius +/- motif; in Figure 10 it also includes what I think is an identifier for the set it comes from; and in Figure 12 it is TRBV + TRBJ + CDR3 + radius. As this is the main focus of the paper I think this definition should be made clear, consistent and explained somewhere.

The nomenclature for defining meta-clonotypes is specified in Supplemental Table 1G and 1H:

Set Name+Background Frequency+ Centroid V-Gene+Centroid CDR3+TCRdist Radius+MOTIF Constraint

(e.g., M_1_1E6+TRBV27*01+CASSDRGPPDTQYF+22+([AST][DE]RG[GP].[DE]TQ) )

– Set Name (e.g., M_1)

– Background Frequency Estimate Used to Set Radius (e.g, 1E6)

– Centroid V-Gene (e.g, TRBV27*01)

– Centroid CDR3 (e.g., CASSDRGPPDTQYF)

– TCRdist Radius (e.g., 22)

– MOTIF Constraint (e.g., ([AST][DE]RG[GP].[DE]TQ))

The reviewer is correct for Figure 10 we do include the meta-clonotype nomenclature: M_55_1E6+TRBV28*01+CASSLKTDAYEQYF+20+(SL[RK][ST][ND].YEQ)

We have addressed the reviewer’s comment and made the identifiers consistent in the updated Figure 12. We have listed identifiers as TRBV27*01+CASSDRGPPDTQYF+22+([AST][DE]RG[GP].[DE]TQ). Because all of these come from a single MIRA set we omit the M_55_1E6 portion of the tag for visual clarity.

Secondly, it was unclear to me when the authors use which set for background: e.g. does Figure 1 and caption 'synthetic set' refer to the set of 100k OLGA V-J biased + 100k cord used later? cf Lines 535-537 "background CDR3s that were sampled from cord blood and constrained to use the same V and J genes", and elsewhere unadjusted cord blood is used without the OLGA adjusted set.

Thank you for the careful review. See comment above about clarifying the captions to explicitly explain the background in each figure. Addressing the previous sentences at line 398-400:

“The background-adjusted plot shows the position-specific Kullback-Leibler divergence from an alignment of background CDR3s that were sampled from cord blood and constrained to use the same V and J genes”

The source of confusion is that the backgrounds used to estimate an optimal meta-clonotype radius are based on V-J gene usage of each of the set of antigen-annotated TCRs. This includes the 200K TCRs: 100,000 OLGA V-J matched combined with a random 100,000 sampled from cord blood.

Whereas for background-adjusted logo motifs and computation of position specific Kullback-Leibler divergence, we use TCRsampler to randomly select TCRs from cord blood with V-J gene usage matching that of the TCRs used for the motif. The new caption states:

“The background-adjusted logos are constructed by randomly sampling TCR β receptors from cord blood with the same TRBV and TRBJ gene usage, with 100 V-J matched TCRs sampled for every receptor in the foreground set.”

The methodology to select a 200K synthetic background is more extensive than that used to generate a background to visualize CDR3 logo motif. More sequences and sampling adjustments are needed to estimate the optimal radius lengths for a large set of antigen-annotated TCRs. By contrast 1000 V-J matched clones is sufficient for the CDR3 logo visualization.

I also think that the advantage / unique usage of the proposed meta-clonotype method over existing methods for grouping antigen-specific TCRs should be further clarified and emphasised: I personally don't find the Results section 'Comparison to k-mer based CDR3 features' or Figures 11 and 12 very convincing to this effect, and think this message should be made clearer in the paper before the sentence in the discussion "Our framework is designed for a different task than these algorithms…".

We have clarified in the Discussion the differences in the design of GLIPH2 and other tools compared to the meta-clonotype framework.

I do think the meta-clonotype method is useful, these changes are addressable and the paper has the potential to be made more readable and more impactful, if the authors streamline the definition and explanations, remove repetitive sections, and cross-reference to the relevant places in the text where things are referred to in the text before their explanations.

Thanks, for your feedback. See above for the changes we’ve made to the Introduction and Discussion which address several of these issues.